# Materials informatics for the screening of multi-principal elements and high-entropy alloys

J.M. Rickman[1,2], H.M. Chan[2], M.P. Harmer[2], J.A. Smeltzer[2], C.J. Marvel[2], A. Roy[3] & G. Balasubramanian[3]

The field of multi-principal element or (single-phase) high-entropy (HE) alloys has recently seen exponential growth as these systems represent a paradigm shift in alloy development, in some cases exhibiting unexpected structures and superior mechanical properties. However, the identification of promising HE alloys presents a daunting challenge given the associated vastness of the chemistry/composition space. We describe here a supervised learning strategy for the efficient screening of HE alloys that combines two complementary tools, namely: (1) a multiple regression analysis and its generalization, a canonical-correlation analysis (CCA) and (2) a genetic algorithm (GA) with a CCA-inspired fitness function. These tools permit the identification of promising multi-principal element alloys. We implement this procedure using a database for which mechanical property information exists and highlight new alloys having high hardnesses. Our methodology is validated by comparing predicted hardnesses with alloys fabricated by arc-melting, identifying alloys having very high measured hardnesses.

[1] Department of Physics, Lehigh University, Bethlehem, PA 18015, USA. [2] Department of Materials Science and Engineering, Lehigh University, Bethlehem, PA 18015, USA. [3] Department of Mechanical Engineering and Mechanics, Lehigh University, Bethlehem, PA 18015, USA. Correspondence and requests for materials should be addressed to J.M.R. (email: jmr6@lehigh.edu)

The study of high-entropy (HE) (or multi-principal element) alloys, typically comprising five or more elements, is a relatively new area of materials research that has attracted intense interest in recent years[1–6] as, in many cases, these systems possess unexpected and superior mechanical properties relative to those of conventional alloys[7–10] as well as enhanced oxidation resistance and magnetic properties[11,12]. In general, these alloys exhibit crystal structures and phases that are believed to be entropically stabilized due to the large number of elements present. In this regard, the early work by Yeh and co-workers[6,13] revealed that a veritable cocktail of metallic elements resulted in a far greater degree of solid–solid solubility than is achievable in mixtures comprising fewer components. Despite substantial progress in this area, however, there remain significant challenges in understanding the origin of these superior properties from a fundamental point of view[14]. For example, the role of complex strengthening mechanisms, including solid–solution strengthening, in determining measured alloy hardness is still a subject of vigorous debate.

It is therefore of considerable interest to identify element combinations and associated compositions leading to high-strength, high-hardness alloys. However, given a large palette of possible elements, the number of potential HE alloys that may be fabricated is exceedingly large. Unfortunately, only a relatively small subset of these alloys is expected to have desirable properties. Thus, to obviate fruitless experimentation in the search for this desirable subset, it is imperative to identify, a priori, candidate systems that are likely to have a high degree of solid–solid solubility and enhanced thermomechanical properties.

In recent years, several groups have devised computational strategies to identify and characterize HE alloys[15–17]. For example, Troparevsky et al.[17] used "high-throughput" density-functional theory calculations of formation energies to predict which combination of elements is likely to form an HE alloy. In addition, Senkov et al.[18] employed a calculated phase diagram (CALPHAD)-based combinatorial approach to screen computationally a large number of candidate metal alloys for those forming only solid–solution phases. Beyond these studies, other workers have formulated predictive metrics, such as average melting temperature and average valence electron concentration, to characterize relatively simple correlations between the metrics and the propensity to form HE alloys. Most recently, Sarker et al. proposed using a new descriptor, the entropy-forming ability, to predict HE alloys having high hardnesses and validated its use by combining first-principles and experimental synthesis[19]. While quite useful, these methods have not really explored the complex inter-relationships among many metrics and the thermomechanical properties of HE alloys nor the use of this information to accelerate alloy design.

With this in mind, we describe here the use of data analytics to accelerate the discovery of new, useful multi-component alloys by determining maximal correlations among metrics and alloy properties in experimental databases and the exploitation of this information to search for and identify promising HE alloy candidates. In particular, by combining a multiple regression analysis or its generalization, a canonical correlation analysis (CCA), with a genetic algorithm (GA) optimization strategy, we will explore systematically the HE alloy chemistry/composition space to highlight those alloys that have beneficial mechanical properties (e.g., high hardness). A CCA is a very general technique for quantifying relationships between two sets of variables, most parametric tests of significance being essentially special cases of CCA[20], and it has been used recently to: explore, for example, correlations between ceramic powder chemistry and the resulting microstructure of a dense, sintered ceramic[21,22]. In this context, a GA is employed to construct many virtual candidate alloys that evolve from one generation to the next by processes that mimic reproduction and mutation, and in which survival to the next generation is dependent upon a measure of fitness. The fitness measure governing alloy selection will be determined from the aforementioned multiple regression/CCA results. This computational approach allows one to screen hypothetical alloys relatively quickly, thereby accelerating the identification and design of new alloy systems. We implement this procedure using a database comprising 82 HE alloys for which reliable mechanical property information is available to highlight new alloys having potentially high hardnesses.

## Results

**Metrics.** We began by compiling a list of $M = 82$ experimentally fabricated HE alloys for which there are measured (Vicker's) hardnesses[8,10,13,23–35]. For each system, we also used (or computed) the values of alloy metrics that have been employed in previous studies to predict the potential for high solid solubility. In particular, for each element $i$ having an atomic radius of $r_i$, melting temperature, $(T_m)_i$, Young's modulus, $E_i$, and valence electron concentration, $\text{VEC}_i$, we considered the following quantities calculated in terms of the molar compositions, $c_i$, for each of the $N$ constituent elements:

Radius asymmetry, $\delta = \sqrt{\sum_{i=1}^{N} c_i \left(1 - \frac{r_i}{\bar{r}}\right)^2}$

Enthalpy of mixing, $\Delta H_{mix} = 4 \sum_{i=1, j \neq i}^{N} (\Delta H_{mix})_{ij} c_i c_j$

Ideal entropy of mixing, $\Delta S_{mix} = -R \sum_{i=1}^{N} c_i \ln c_i$

Mean melting temperature, $\bar{T}_m = \sum_{i=1}^{N} c_i (T_m)_i$

Entropy/enthalpy ratio, $\Omega = \frac{\bar{T}_m \Delta S_{mix}}{|\Delta H_{mix}|}$

Young's modulus asymmetry, $\varepsilon = \sqrt{\sum_{i=1}^{N} c_i \left(1 - \frac{E_i}{\bar{E}}\right)^2}$

Valence electron concentration, $\text{VEC} = \sum_{i=1}^{N} c_i \text{VEC}_i$

In these expressions, the bar denotes the composition-weighted mean, $(\Delta H_{mix})_{ij}$ is the regular solution enthalpy of mixing associated with elements $i$ and $j$ calculated from Miedema's model[36], and $R$ is the gas constant.

**Parallel coordinate plot and correlation analysis.** It is of interest to identify significant relationships between the alloy metrics (i.e., predictor variables) and the corresponding thermomechanical properties (i.e., outcome variables) of the alloy. For this purpose, four such outcomes were examined, namely: (1) the presence of body-centered cubic (bcc) solid solution(s) only, (2) the presence of face-centered cubic (fcc) solid solution(s) only, and (3) the presence of intermetallic (IM) phase(s), and (4) the Vicker's hardness ($H$). To establish a basis for comparison, it is useful to relate both predictor and outcome variables to those of a reference alloy, conveniently chosen here to be CoCrFeNiCu, and to denote the ratio of a given variable to that of a reference variable with a prime. This choice was made given that the reference alloy is relatively well studied. Figure 1 summarizes these normalized metrics and outcomes for the 82 HE alloys in the form of a parallel coordinate plot. This type of plot replaces the conventional $d$-dimensional orthogonal Cartesian axes by a set of $d$ parallel axes, with each $d$-dimensional point represented by a polyline in parallel coordinates, and is very useful for displaying high-dimensional data[37–39] and identifying some global trends. As is evident from the figure, there does not appear to be any strong correlation between phase behavior and hardness, whereas, for a subset of alloys, there is some correlation between valence electron concentration and hardness.

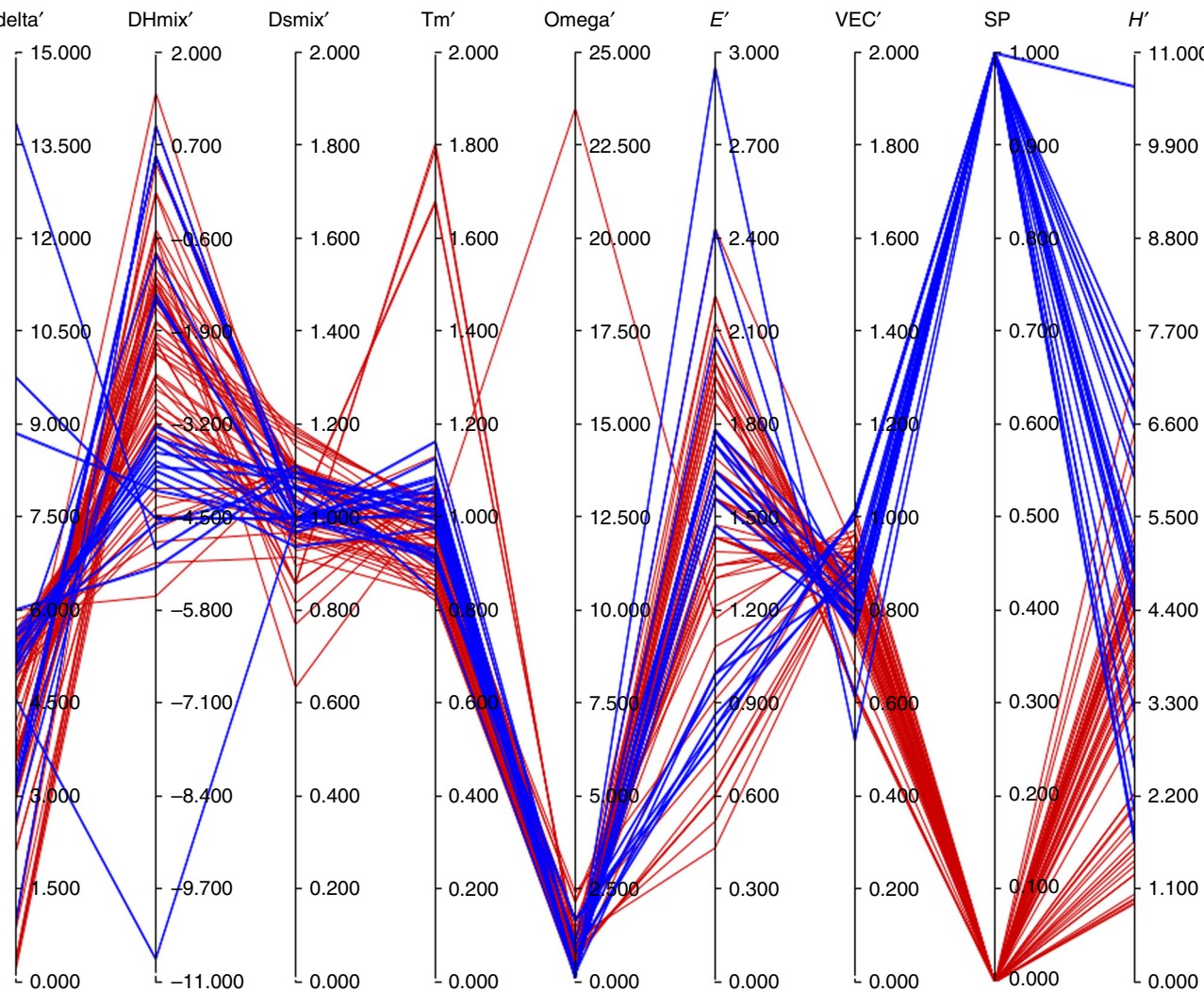

**Fig. 1** Parallel coordinate plot. A parallel coordinate plot displaying both normalized metrics and property data for a set of high-entropy (HE) alloys. The phase behavior of the alloys is summarized in the variable single phase (SP) (unnormalized), with SP = 1 indicating alloys comprising a single, solid–solution (either fcc or bcc) and SP = 0 otherwise. Solid–solution alloys are shown in blue, with the remaining alloys having intermediate phases, etc., shown in red

To be more quantitative, we employ a multiple regression analysis to assess the relative importance of various materials characteristics whose complex interplay dictates properties. Our analysis begins with a vector of predictor variables, $x$, and outcome variables, $y$, taken as

$$x = \{\delta', \Delta H'_{\mathrm{mix}}, \Delta S'_{\mathrm{mix}}, T'_{\mathrm{m}}, \Omega', \varepsilon', \mathrm{VEC}'\},$$
$$y = \{H'\},$$

$$(1)$$

where $y$ consists of only $H'$ for a multiple regression. (A more general CCA analysis that is useful when there are additional outcome variables in $y$ is described below in the "Discussion" section). It should be noted that correlations, some of which are significant, exist among the predictor variables as can be seen by inspection of a color map of the correlation matrix (Fig. 2) given below.

From this analysis, the correlation coefficient was found to be $0.79 \pm 0.07$ with $p \approx 0$. An examination of the resulting canonical weights (i.e., the $\alpha_i$ summarized in Table 1) reveals that three of the predictor variables, namely $\Delta S'_{\mathrm{mix}}$, $\Delta H'_{\mathrm{mix}}$, and VEC', are especially relevant in determining the hardness.

More specifically, from Table 1 it can be seen that $\Delta S'_{\mathrm{mix}}$ are positively correlated with the $H$, suggesting that compositional disorder enhances the hardness. By contrast, $\Delta H'_{\mathrm{mix}}$ and VEC' are negatively correlated with the hardness, suggesting in the first case that phase separation is associated with lower hardness values (HV). In the latter case, the negative correlation of VEC' with hardness is consistent with recent findings that a low VEC is associated with an improvement in strength via the promotion of a bcc phase[40].

Finally, one may ask whether the three principal metrics, $\Delta S'_{\mathrm{mix}}$, $\Delta H'_{\mathrm{mix}}$, and VEC', embody most of the physics that determines the hardness. If so, one would expect that an analysis based only on this reduced set of variables would satisfactorily predict the hardness. To assess the adequacy of this (null) hypothesis, we conducted a correlation analysis of this reduced model and computed the resulting $F$-statistic that compares the sum-of-squares errors for the reduced and full models[41]. The resulting $p$-value of 0.81 suggests that the reduced model is indeed adequate. It should be emphasized, though, that further microstructural (and possibly computational) analysis is needed to confirm these statistical insights.

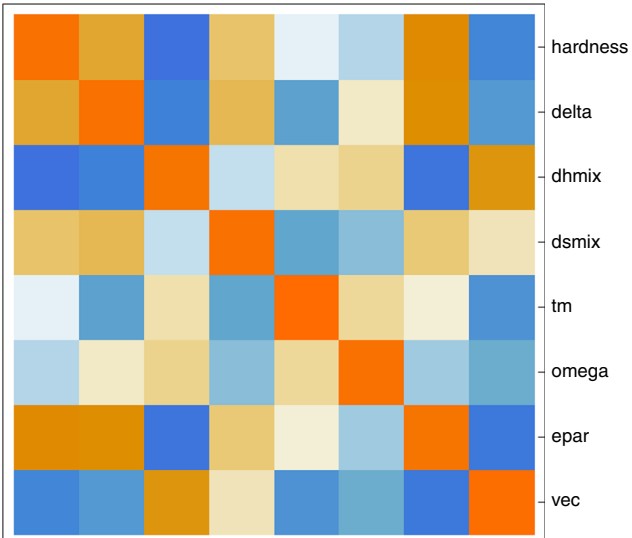

**Fig. 2** Correlation matrix map. A color map of the correlation matrix, with both alloy metrics and the hardness labeled. The color scale ranges from red to blue, with dark red indicating a correlation of $+1$ and dark blue a correlation of $-1$. No correlation (0) corresponds to white

### Table 1 The standardized canonical weights found in the CCA with hardness as the sole output variable

| $\delta'$ | $\Delta H'_{mix}$ | $\Delta S'_{mix}$ | $T'_m$ | $\Omega'$ | $\varepsilon'$ | $VEC'$ |
|---|---|---|---|---|---|---|
| $-0.047$ | $-0.553$ | $0.285$ | $-0.095$ | $-0.108$ | $0.057$ | $-0.476$ |

*CCA* canonical correlation analysis, *VEC* valence electron concentration

Figure 3 shows the values for the correlated variates, $V^{(1)}$ and $W^{(1)} = (H' - \langle H' \rangle)/\sigma$, for each of the $M$ alloys and, in addition, a regression line that highlights the relationship between the variates. (The prime here denotes a value normalized by the reference alloy, the angle brackets denote an average over a dataset comprising the $M = 82$ HE alloys, and $\sigma$ denotes the standard deviation in $H'$ over the dataset). Despite the adequacy of the reduced model, for greater accuracy the multiple regression analysis for the full model will be used below to identify promising alloys.

**Alloy identification using a GA.** Having obtained canonical variates from the above analysis, the variate $V^{(1)}$ was used as a fitness function in a GA to find candidate alloys having high hardness. While it may be possible to narrow the search space to some degree by analytical optimization, the non-linear dependence of some of the alloy metrics on composition suggested a numerical solution using all of the metrics. More specifically, to construct candidate, 5-element alloys, a 16-element palette and 16 molar compositions per element were employed to represent $\binom{16}{5}(16)^5$ independent alloys. The alloys in this palette are Co, Cr, Fe, Ni, Al, Cu, Mn, Ti, Mo, Nb, Ta, V, W, Zr, Zn and Sn.

The calculation began with $N_c = 500$ randomly selected chromosomes, each a 40-bit string encoding the chemistry and composition of an alloy having distinct elements, with the $i$th chromosome having a fitness $f_i$. Successive generations were produced with a series of evolutionary processes, including fitness selection, recombination, and mutation. In particular, from each

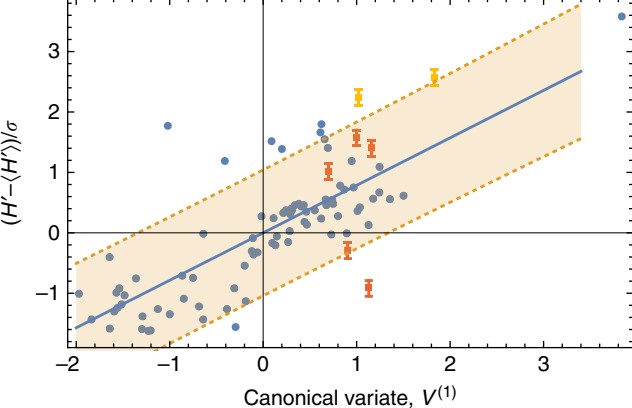

**Fig. 3** Correlation analysis. The canonical variate, $V^{(1)}$, versus the canonical variate, $W^{(1)} = (H' - \langle H' \rangle)/\sigma$, for each alloy (blue circle) and the corresponding regression line (solid blue). The Vicker's hardness is $H$ and $\sigma$ denotes the standard deviation in normalized hardness. The dotted lines delimit a shaded 90% (single observation) prediction band. Also shown are the measured (normalized) micro-hardnesses for 7 fabricated alloys (5 orange squares, with the 2 alloys having hardnesses in excess of 1000 hardness values (HV) in yellow.) The error bars were estimated from approximately 20 separate hardness measurements per sample. The standard error of the estimate, as obtained using the data for the synthesized alloys, is approximately 259.2 HV

generation, $N_c$ chromosomes were selected with probability $p_i = f_i / \sum_j f_j$ for the $i$th chromosome. (Thus a given chromosome may be, and often is, chosen more than once.) These selected chromosomes were then subjected to a recombination process in which pairs are chosen with probability $p_r = 0.05$, and then, upon randomly selecting bits, chromosomal strands are exchanged between a given pair. Finally, a mutation process was modeled in which randomly selected bits, chosen with probability $p_m = 0.001$, were changed from 0 to 1, or vice versa. If any of these processes resulted in a chromosome with one or more repeated elements, a new chromosome comprising distinct elements was generated. It should be noted that some tuning of the parameters $p_r$ and $p_m$ was required to create an algorithm in which successive generations have markedly improved fitness distributions. In practice, one observes the evolution of the fitness function over successive generations and then adjusts the parameters so that there is substantial improvement over many generations. Multiple trials consisting of at least 1000 generations were used to identify superior alloys.

The utility of these may also depend, at least in part, on their ductility. Given the inherent tradeoff between hardness and ductility in most systems, it is useful to also screen for ductility in this procedure. This may be accomplished via the implementation of a multi-objective (Pareto) optimization GA in which a composite objective function is formulated from a weighted sum of two objective functions, namely, one for the hardness (as obtained from the canonical variate calculation) and one for the ductility[42]. As a proxy for ductility measurements, it is useful to employ the empirical Pugh criterion[43] that correlates ductility with the ratio $P = \mu/B$, where in this context $\mu$ is the composition-weighted mean shear modulus and $B$ is the corresponding bulk modulus. Ductile behavior is then associated with $P < 0.5$. In this work, we have chosen to simply evaluate $P$ for promising candidates, as summarized below.

Figure 4 displays the cumulative distribution of fitnesses both at the beginning of the GA and after 1000 generations. As is evident from the figure, the population of alloys better matches

the hardness criterion after many iterations. From these simulations, we identified those alloys that are associated with the largest fitness values as candidates for high-hardness alloys.

**Comparison with synthesized alloys**. To assess the predictive capability of our methodology, we fabricated and hardness tested seven candidate alloys that were identified as promising from the aforementioned GA. Given that the GA produces an extremely fit population of high-hardness alloys that represent at optimality an extreme extrapolation from the $M = 82$ dataset, we conservatively select from this population a subset of alloys whose predicted hardnesses are somewhat beyond the range of the existing dataset (i.e., having fitness values that are about 10–15% greater than the maximum found with the multiple regression). Our view is that, used in this way, the GA systematically creates a large pool of potential candidates having relatively high hardnesses for a range of fitness values that represents a conservative extrapolation of the data. Parameter tuning in the GA is then performed to create this large pool. This extrapolation can be tested step by step, and if desired, the new alloys discovered in this way may be then added to the existing data to bootstrap the extrapolation. The candidate HE alloys were synthesized by arc-melting compressed pellets of elemental, high-purity powders (Sigma-Aldrich, purity ≥99.9%). (Powdered metals were used to limit the occurrence of macro-segregation of elements and to improve homogeneity. Also, arc-melted pellets were remelted four times to ensure homogeneity.) The resulting as-cast, solid pieces were then subjected to micro-hardness testing with a 100 g load to determine their respective Vicker's HV, the values taken to be the mean of 20 independent tests per piece.

The resulting scaled hardnesses are also displayed in Fig. 3 as a function of the canonical variate, $V^{(1)}$. It can be seen from the figure that about 5 out of 7 of the hardnesses are within the pictured 90% prediction interval. The highest measured hardnesses were found for $Co_{33} W_{07} Al_{33} Nb_{24} Cr_{03}$ (1084 ± 37 HV) and $Ti_{18} Ni_{24} Ta_{12} Cr_{22} Co_{24}$ (1011 ± 20 HV) that have corresponding Pugh ductility ratios of $P = 0.38$ and $P = 0.47$, respectively (For comparison, the Vicker's HV for alloys such as Ti-5Al, Ti-5V, and 75Au-7Cu-13Ag-5Co (wt. %) are 532.3, 429.3 and 128, respectively[44,45]). Indeed, for each of the 7 candidate alloys identified here, $P < 0.5$. (However, for the hardest alloy a close inspection of the hardness indentations did reveal some degree of brittleness as cracking occurred near indentation corners in addition to observable cooling cracks.) The HV for the 7 synthesized alloys are summarized in the table below.

In summary, given the number of candidate alloys considered here and the inherent scatter in the original data, this analysis provides a convenient means to predict alloys with enhanced hardness with 90% confidence.

**Microstructural and microchemical characterization**. To understand better the observed plastic response of the synthesized alloys, the microstructure of the samples with compositions $Co_{33} W_{07} Al_{33} Nb_{24} Cr_{03}$ and $Ti_{39} W_{04} Nb_{31} Ta_{04} Co_{22}$, the alloys having the highest and lowest HV (see Table 2), respectively, were examined using scanning electron microscopy (SEM). Figures 5 and 6 depict the microstructure of $Co_{33} W_{07} Al_{33} Nb_{24} Cr_{03}$ using secondary electron intensity, as well as the associated compositional maps of the constituent elements obtained by X-ray energy-dispersive spectroscopy (EDS). It can be seen that the microstructure is dendritic (which is consistent with a cast alloy), with a relatively small volume fraction (13 vol. %) of inter-dendritic phase. The EDS data show that the dendritic phase is enriched in Nb and W. Given that there is no obvious partitioning of the Nb and W, it is assumed that they form a solid solution. This behavior would not be unexpected given that both Nb and W are bcc and exhibit complete solid solubility[46]. The SEM also reveals that the inter-dendritic phase exhibits a two-phase, eutectic-like morphology. Of the two phases, one is Nb rich, whereas the other contains higher proportions of Al, Co, and Cr, with little Nb. Unlike for the case of the dendrite body, it does not appear that there is strong preferential association of the W with the Nb-rich component.

**Table 2 The Vicker's hardness values (in HV) for the synthesized alloys**

| Alloy | Hardness (HV) | Hardness (HV) predicted |
|---|---|---|
| $Co_{33} W_{07} Al_{33} Nb_{24} Cr_{03}$ | 1084 ± 37 | 825 |
| $Ti_{18} Ni_{24} Ta_{12} Cr_{22} Co_{24}$ | 1011 ± 20 | 677 |
| $Co_6 W_9 Al_{36} Mo_{38} Ni_{11}$ | 725 ± 47 | 618 |
| $Ni_{47} Co_{02} Ta_{12} Ti_9 Nb_{30}$ | 815 ± 43 | 702 |
| $Ti_{44} Ni_{02} Nb_{21} Cr_{21} Co_{12}$ | 422 ± 13 | 656 |
| $Ti_{32} Nb_9 Ta_{01} Cr_{19} Co_{39}$ | 856 ± 29 | 673 |
| $Ti_{39} W_{04} Nb_{31} Ta_{04} Co_{22}$ | 277 ± 12 | 697 |

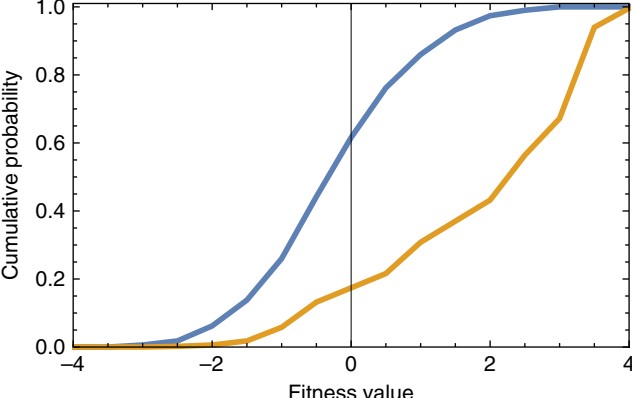

**Fig. 4** Genetic algorithm (GA) fitness distribution. The cumulative distribution of fitnesses both at the beginning (shown in blue) of the GA and after 1000 generations (shown in orange). Clearly, the alloy distribution better matches the hardness criterion after many iterations

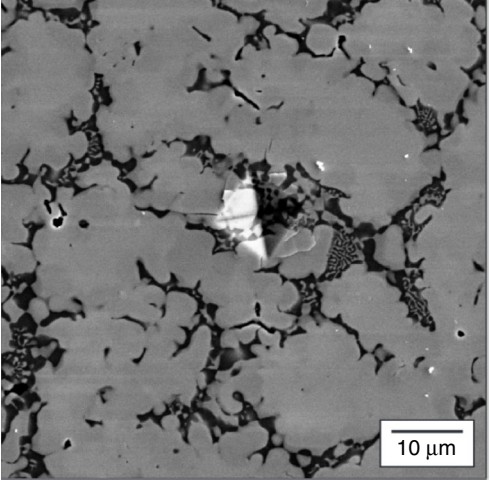

**Fig. 5** Micrograph of a hard alloy. A scanning electron microscopic micrograph in a region near a microhardness indent in the $Co_{33} W_{07} Al_{33} Nb_{24} Cr_{03}$ sample that was synthesized by arc-melting. The figure evinces a dendritic microstructure with eutectic-like inter-dendritic regions

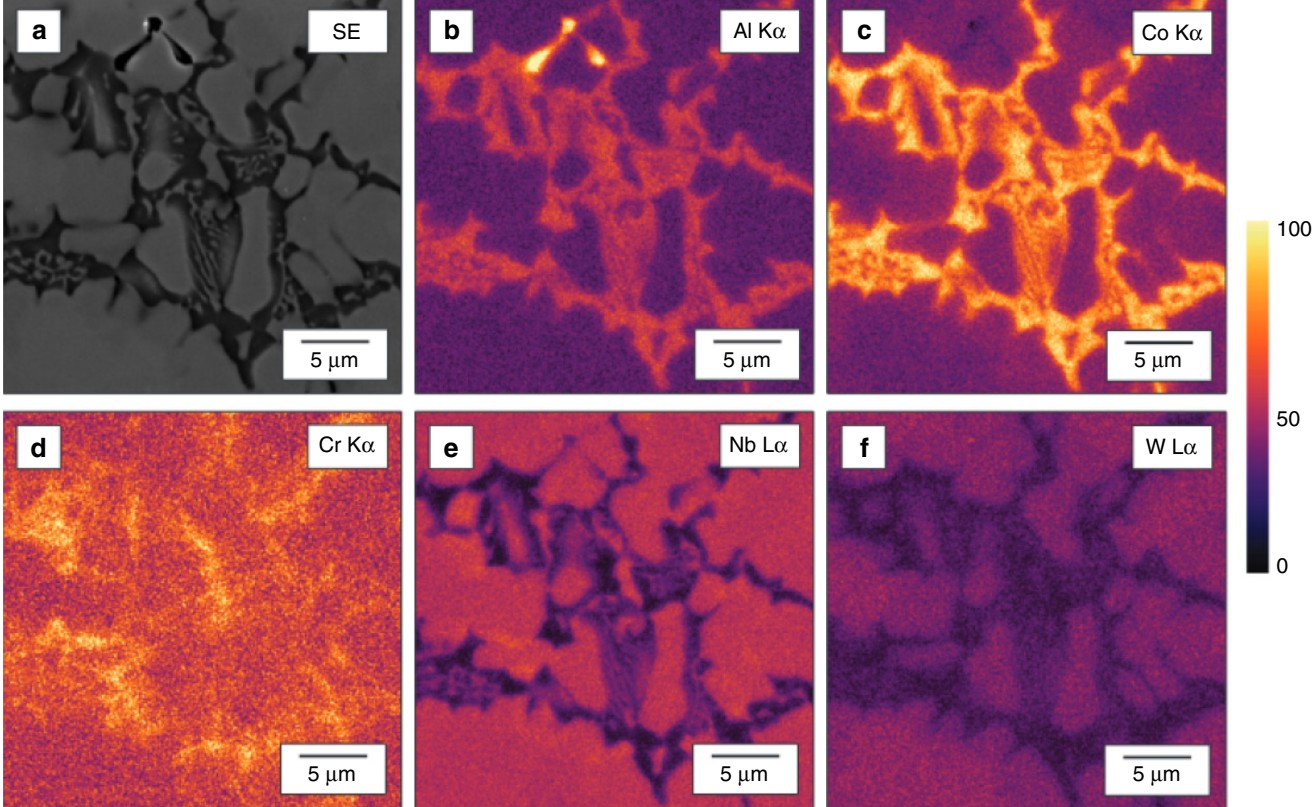

**Fig. 6** Energy-dispersive spectroscopic (EDS) map of a hard alloy. An X-ray EDS map of the as-cast microstructure of $Co_{33} W_{07} Al_{33} Nb_{24} Cr_{03}$.
**a** Secondary electron image and X-ray intensity maps produced by integrating **b** Al Kα, **c** Co Kα, **d** Cr Kα, **e** Nb Lα, and **f** W Lα X-ray signals. The color scale represents the relative intensity based on the number of counts per pixel. The map suggests that the dendritic region of the microstructure is rich in Nb and W while the eutectic inter-dendritic region comprises two phases, one rich in Nb and the other primarily composed of Al, Co, and Cr

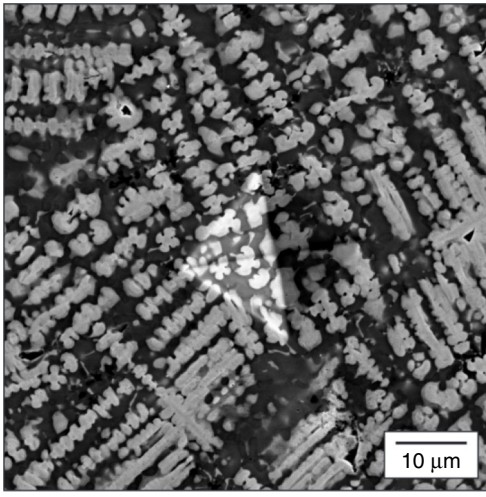

**Fig. 7** Micrograph of a soft alloy. A scanning electron microscopic micrograph in a region near a microhardness indent in $Ti_{39} W_{04} Nb_{31} Ta_{04} Co_{22}$ in a sample that was synthesized by arc-melting. The figure shows a dendritic microstructure similar to that of traditionally cast alloys

The corresponding images for the softest alloy are shown in Figs 7 and 8. This microstructure of $Ti_{39} W_{04} Nb_{31} Ta_{04} Co_{22}$ is also dendritic, but the volume fraction of inter-dentritic material is much higher here (53 vol. %). In this alloy, the dendrites are enriched in Nb, Ta, and W, whereas the inter-dendritic regions contain a greater proportion of Co and Ti. In the SEM image

(Fig. 7), the slight differences in contrast at the edges of the dendrites suggest the possibility of coring. This observation is confirmed by the compositional maps where it can be seen that, for a given dendrite structure, the spatial extent of Nb and Ta enrichment is the greatest while W is confined to the dendrite inner core (see, in particular, Fig. 8g). Thus one infers that, although all three elements are present in the dendritic regions, the spatial distribution is inhomogeneous.

The marked difference in hardness between the two alloys is very interesting, particularly given that the compositional make-ups of the dendritic phases are not that dissimilar. In both cases, there is a significant component of Nb and W. In the case of the softest alloy, Ta is also present. One possible explanation is that the high hardness of $Co_{33} W_{07} Al_{33} Nb_{24} Cr_{03}$ is attributable to the significant volume fraction of the Nb–W phase, which is solid solution strengthened. One could suggest that the same degree of strengthening is not present in $Ti_{39} W_{04} Nb_{31} Ta_{04} Co_{22}$ either due to the non-uniform distribution of the elements or the additional presence of Ta. Of course, the potential contribution of alloying elements present at relatively low concentrations cannot be excluded.

Finally, we have also assessed the ductility of each synthesized alloys by optical microscopic inspection of the (20) microhardness indentations per alloy for signs of cracks at the corners of the indentations. It was found that 2 of the 7 systems showed moderate to considerable ductility (i.e., $\gtrsim$50% of the indentations without cracks), including the relatively hard alloys $Ti_{32} Nb_9 Ta_{01} Cr_{19} Co_{39}$ and $Co_6 W_9 Al_{36} Mo_{38} Ni_{11}$, while the hard alloy $Ti_{18} Ni_{24} Ta_{12} Cr_{22} Co_{24}$ showed limited ductility (i.e., ≈15–20% of the indentations without cracks). In addition, the relatively low hardness alloys, $Ti_{39} W_{04} Nb_{31} Ta_{04} Co_{22}$ and $Ti_{44} Ni_{02} Nb_{21}$

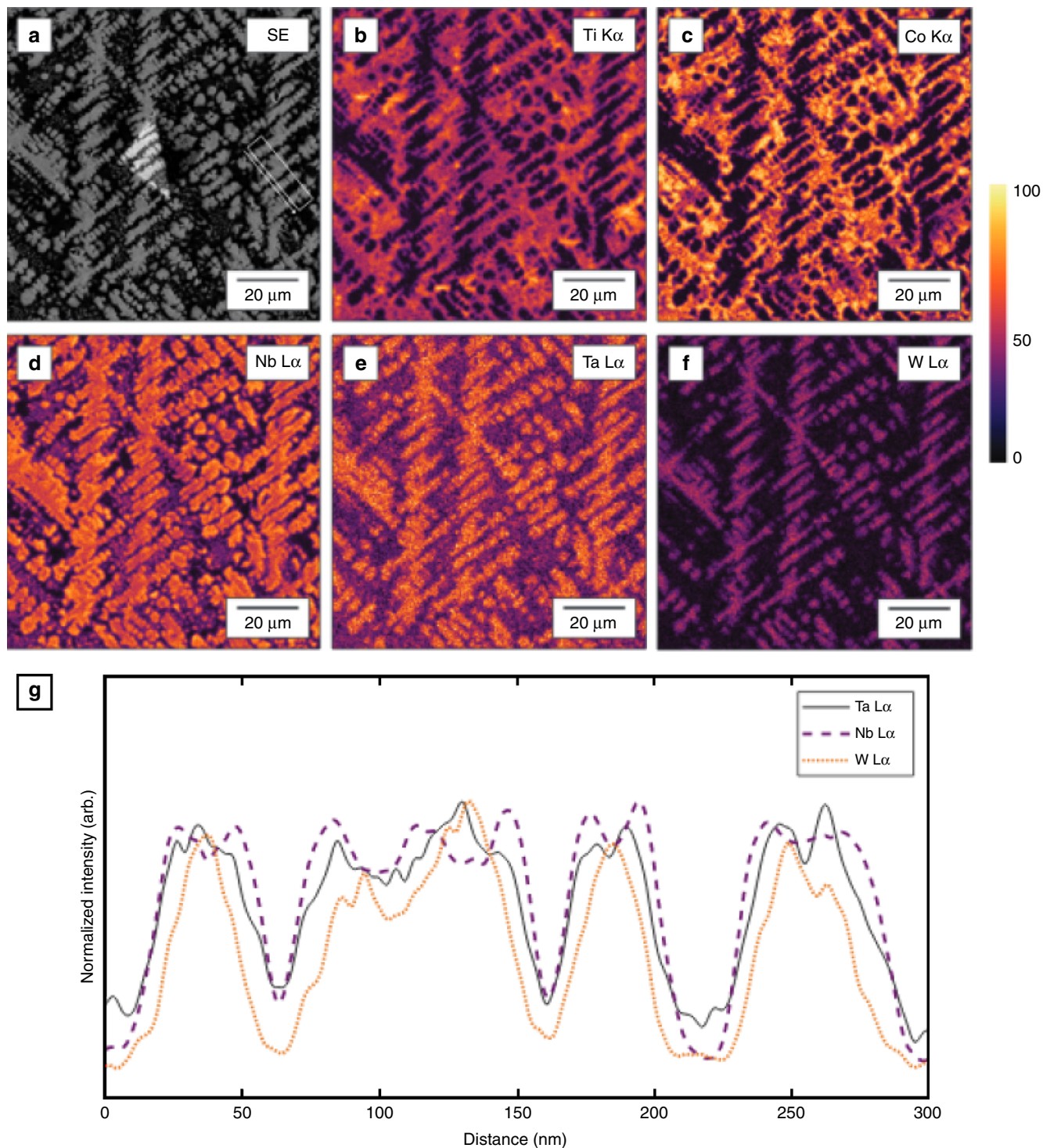

**Fig. 8** Energy-dispersive spectroscopic (EDS) map of a soft alloy. An EDS map of the as-cast microstructure of $Ti_{39} W_{04} Nb_{31} Ta_{04} Co_{22}$ and evidence of coring. **a** Secondary electron image and X-ray intensity maps produced by integrating **b** Ti K$\alpha$, **c** Co K$\alpha$, **d** Nb L$\alpha$, **e** Ta L$\alpha$, and **f** W L$\alpha$ X-ray signals. The color scale represents the relative intensity based on the number of counts per pixel. The dendrites were found to be rich in Nb, Ta, and W, while the inter-dendritic phase is composed primarily of Co and Ti. Evidence of dendrite coring can be seen in **g**, a line profile showing the normalized intensity of W L$\alpha$, Ta L$\alpha$, and Nb L$\alpha$ X-ray signals taken from the box in **a** with an arrow indicating the direction of the line profile from left to right

$Cr_{21} Co_{12}$, were found to be quite ductile (i.e., $\gtrsim 95\%$ of the indentations without cracks), which is generally consistent with observations that Group IV element additions, such as Ti, are often associated with increased ductility[47]. The remaining alloys were found to be brittle. Given the empirical nature of the Pugh criterion, it is perhaps unsurprising that it is a rough predictor of ductility in

this context. Nevertheless, with this methodology we have identified several alloys that are both hard and relatively ductile.

## Discussion

We present here a data analytics approach to screen efficiently multi-principal component alloys using a multiple regression

analysis and a GA. Our methodology permits the exploration of the multi-dimensional chemistry/composition space that is associated with these alloys and thereby the identification of alloys having beneficial mechanical properties. In particular, two alloys were discovered with extremely high HV in excess of 1000 HV, and these predictions were validated by micro-hardness testing of alloys that were fabricated via arc-melting. The most promising alloys were also screened to ensure that they possessed reasonable ductility.

Given the analysis described above, it is intriguing to consider the role of minor elements in determining alloy hardness, especially for $Co_{33} W_{07} Al_{33} Nb_{24} Cr_{03}$. Our methodology permits a detailed investigation of the impact of small composition changes in a reduced parameter space containing, for example, these minor elements. We therefore examined the behavior of the fitness function obtained from the regression for small changes in either Cr alone or W alone. It was found that the maximum hardness occurred for a Cr composition close to 0.07 and for a W composition close to 0.04. Thus, for experimental guidance, it is possible to fine tune the predictions made here by considering small compositional perturbations.

The methodology summarized here can, of course, be employed to examine the influence of the predictor set on hardness and phase behavior, as summarized in Eq. (2) below. This analysis is useful if one wishes to obtain hard alloys that are also likely to be a single solid solution. To perform this analysis, one employs a CCA with an extended set of output variables, $\vec{y}$, (see below) for the calculations of canonical weights and variates. Moreover, the CCA and GA methodologies are quite general and can therefore be applied to properties other than hardness. As noted in the "Introduction" section, a CCA and its relative, a Monte Carlo CCA, have been used in a very different context to establish correlations between ceramic powder chemistry and the resulting microstructure of a dense, sintered ceramic. A similar CCA analysis has also been used to relate the microstructure to the optoelectronic properties of thin-film solar cells[22]. Thus, in this context, the CCA can also be generalized to identify alloys having other useful thermomechanical and kinetic properties (e.g., yield strength, electrical conductivity, corrosion resistance) or, as described above, to Pareto-optimize multiple properties simultaneously using a multi-objective GA. Moreover, this approach can also be employed to investigate the impact of processing on property measurements. For example, recent work suggests that a nominally single-phase, HE alloy, namely, $Mo_{25} Nb_{25} Ta_{25} W_{25}$, synthesized by mechanical alloying comprised, in fact, multiple phases. Its associated complex microstructure, along with any subsequent annealing of the system, may dictate, at least in part, its measured mechanical properties (Smeltzer, J. A. unpublished work (2018)). The investigation of such processing/property correlations is the subject of ongoing work.

Finally, to illustrate the use of the CCA to examine multiple output variables, we consider the following set of predictor ($x$) and output ($y$) variables, respectively, for the HE alloy example.

$$x = \{\delta', \Delta H'_{\mathrm{mix}}, \Delta S'_{\mathrm{mix}}, T'_{\mathrm{m}}, \Omega', \varepsilon', \mathrm{VEC}'\},$$
$$y = \{\mathrm{fcc}, \mathrm{bcc}, \mathrm{IM}, H'\}, \quad (2)$$

where fcc, bcc, and IM denote the presence of a fcc solid solution, a bcc solid solution, or intermediate phase(s), respectively. One then finds linear combinations (known as canonical variates) $V = \sum_i \alpha_i x_i$ and $W = \sum_i \beta_i y_i$, where the coefficients $\alpha_i$ and $\beta_i$ are called canonical weights, such that these combinations are maximally correlated. For this purpose, one constructs the associated correlation (or covariance) matrix, $\Sigma$, and defines the

operators $\sigma_1$ and $\sigma_2$ from the blocks of $\Sigma$ as[48,49]

$$\sigma_1 = \Sigma_{\vec{x}\vec{x}}^{-1} \Sigma_{\vec{x}\vec{y}} \Sigma_{\vec{y}\vec{y}}^{-1} \Sigma_{\vec{y}\vec{x}},$$
$$\sigma_2 = \Sigma_{\vec{y}\vec{y}}^{-1} \Sigma_{\vec{y}\vec{x}} \Sigma_{\vec{x}\vec{x}}^{-1} \Sigma_{\vec{x}\vec{y}}. \quad (3)$$

The use of the correlation matrix here implies that one is examining relationships among standardized variables[48]. The square roots of the eigenvalues of $\sigma_1$ and $\sigma_2$ are the canonical correlations. Moreover, the eigenvectors corresponding to the maximum eigenvalue are the desired canonical variates that maximize the correlation. We note that the CCA methodology has been extended to identify non-linear variable combinations that are highly correlated[22,50,51].

We performed a CCA using the HE alloy dataset described above. From this analysis, one finds three significant pairs of canonical variates having correlation coefficients 0.82, 0.63 and 0.53, respectively, with associated errors of approximately ±0.06. The significance of these results is determined by employing a hypothesis test in which one tests the null hypothesis that a given pair of variates is uncorrelated via the construction of an appropriate statistic, such as Wilks lambda[52]. The small calculated $p$-values found here are each <0.001, indicating that a correlation between variates exists. From this investigation, one can identify cases in which high hardness is associated with solid solution behavior and other cases in which high hardness is associated with the presence of IM phases. An examination of the variates indicates that the first variate (i.e., the one having the largest correlation coefficient) is most associated with the hardness and accounts for nearly 50% of the variability. The remaining variates primarily reveal associations between the alloy metrics and the phase behavior variables with only a weak dependence on hardness. Thus this general CCA links the alloy metrics with both properties and phase information and is especially useful to workers who distinguish between HE alloys and multi-principal element alloys based on whether the system is a solid solution.

## Methods

**Data analytics**. We employ here a multiple regression analysis and its generalization, a supervised learning strategy known as CCA, and then use the output from this analysis to construct a fitness function as an input to a GA. The aim of the CCA is to identify those linear combinations of predictor variables that are maximally correlated with the outcome variables. This is accomplished by constructing from products of blocks of the correlation (or covariance) matrix an operator whose eigenvalues ($\lambda_i$, where $i$ runs from 1 to the number of variates pairs, $nv$) capture the degree of correlation between combinations of predictor and outcome variables and whose eigenvectors determined the relative weights of the variables[20,22,48]. The correlation coefficient associated with a given variate $i$ is $\sqrt{\lambda_i}$ and the associated variability is $\lambda_i / \sum_{j=1}^{nv} \lambda_j$. In this context, a GA is a fitness function optimizer that begins with a random distribution of chromosomes (i.e., alloys) represented as fixed-length bit strings and, via evolutionary processes, produces successive generations that are better trial solutions to the problem (i.e., having good mechanical properties), as determined from the fitness function[53–55]. Some tuning is required with respect to the frequency of these evolutionary processes, such as recombination and mutation, to achieve a well-functioning algorithm.

**Microstructural and microchemical characterization**. Microstructural characterization was conducted using SEM. Secondary electron imaging and X-ray EDS were performed on a FEI Scios Dual Beam FIB/SEM operated at an accelerating voltage of 20 kV and beam current of 13 nA. Secondary electron images were collected using an Everhart-Thornley detector. X-ray EDS maps were collected using an EDAX Octane Elite detector and exported via the EDAX Team software.

**Hardness testing**. Vicker's indentation hardness testing was conducted on each as-cast alloy using a LECO LM 248AT model hardness tester. Each indent was performed on a polished surface, and a 100 g load was applied for 10 s. At least 20 indents were conducted for each alloy using a 2 × 10 array, and the error was calculated to reflect 95% certainty using the Student's $t$ distribution.

## Data availability

The authors will make available, upon request, the data used in the applications described in this work. It is understood that the data provided will not be for commercial use.

## Code availability

The authors will make available, upon request, the code used in the applications described in this work. It is understood that the code will not be for commercial use.

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

## Acknowledgements

The authors acknowledge support from the Office of Naval Research under Grant Numbers N00014-18-1-2181 and N00014-18-1-2484. They also acknowledge support from the Nano/Human Interface initiative at Lehigh University.

## Author contributions

J.M.R. is the primary author (and guarantor) of this work. He developed the formalism and did most of the analysis. H.M.C., M.P.H. and G.B. contributed substantially to the discussion of the correlation of hardness with alloy metrics. J.S., C.J.M. and A.R. performed the mechanical testing and the microstructural analysis and also contributed to interpretation of the results.

## Additional information

**Competing interests:** The authors declare no competing interests.

