## [Peer Review File · Nature Communications]

Reviewers' comments:

Reviewer #1 (Remarks to the Author):

The paper reports a recent effort aimed at developing "high-entropy" alloys with improved properties (notably higher hardness) based on a supervised statistical learning technique known as canonical correlation analysis (CCA) that is "trained" using a set of experimental hardness data.

My main concern is that it is not clear why the fitness function should be determined by CCA and why CCA is helpful at all in this problem.

In an alloy design problem, one typically has some goals and constraints and those uniquely determine what the optimization problem should be. In this context, all that is needed is a standard statistical regression of each target variable (say hardness) on the various "alloy metrics". No CCA is needed for this.

Also, some of the various "alloy metrics" are fairly simple functions of composition, so it should be possible to substantially narrow down the search by performing some of the optimization analytically rather than entirely by brute force genetic algorithm (GA). It's not really a case where they are a crucial part of the solution.

Finally, one does not get a sense that the proposed alloys are in any sense "optimal". The statement: "Given that the GA produces an extremely fit population of high-hardness alloys that represent (in some cases) an extreme extrapolation from the $M = 82$ dataset, we conservatively select from this population a subset of alloys whose predicted hardnesses are somewhat beyond the range of the existing data set." raises some questions in that regard. (And, no, removing the sentence is not the solution.) What would have been the best candidate materials based on the model? Why exclude it?

Overall, the authors did do a lot of work (on the experimental side) and do suggest improved alloys, so the paper does have merit. It's just a matter of motivating the methodology better or just improving it.

More minor points:

- Figure 1 is not helpful at all.
- Please cite where you obtained these alloy metrics from.

Reviewer #2 (Remarks to the Author):

In the paper "Materials Informatics For the Screening of Multi-Principal Elements and High-Entropy Alloys", Rickman et al. propose a method to find new high-entropy alloys (HEAs) using canonical correlation analysis (CCA) and genetic algorithms (GA). They predict seven new HEAs, two of them having hardness values above 1,000. The idea behind this method appears solid, but there are a few questions about the execution. I do not think there is sufficient data in this manuscript to justify some of their claims, and there does not appear to be sufficient additional physical insights to justify a publication in Nature Communications. I also find the necessity of tuning the GA parameters, especially without further description of how they were tuned, to be problematic. The following is a more detailed critique of their work.

First of all, the description of the currently existing literature on the prediction of HEAs needs to be improved.

The discussion of the CCA analysis could significantly improve by being more explicit. There are a

few choices that the authors make that are unclear. For example, they “conveniently” chose the reference alloy CoCrFeNiCu, but it is unclear to me what is convenient about this choice.

Another such choice is the inclusion of the crystal structure properties (BCC only, FCC only, intermetallic phases) in the outcome variables. Including these variables seems especially superfluous since a) these properties are never discussed and b) in the last paragraph of the CCA analysis, they were omitted anyway. The authors should describe why these outcome variables were included in the first place and how they are relevant to the paper.

On page 7, the authors mention the correlation values of three canonical variates, but do not describe what these variates are or what is different between the variates with different correlations.

The paper then proceeds with using a GA to find solid-solution forming alloys using a select number of elements. Here again, the authors should justify why they selected these particular metals. The GA algorithm helped identify seven promising HEAs, which they then attempted to synthesize. I have several questions and comments about this part. First of all, the method and instrument with which the hardness is determined are neither described nor cited. Second, the authors present no experimental evidence that these compounds are solid solutions, which is what they wanted to find, or that they are homogeneous. At last, I find their presentation of how well experiment and model agree questionable. 70% sounds very good, but it is only based on 7 data points, so the error bar is considerable. Moreover, this value can only be reached by including the data point that just barely touches the 90% confidence interval. I do not think that this is an accurate representation of the predictive power of the model without seeing more data.

The authors also highlight the two compounds with a hardness value of over 1,000 in Figure 2. Interestingly, these are the values that are slightly above the 90% confidence interval. Is there a hypothesis about why that is the case? The authors should discuss this. And what is the source of the remaining outlier?

As the authors mention, the sample size is fairly small due to the limited amount of experimental data. This naturally introduces bias in the dataset. Furthermore, the authors needed to “tune” the mutation and recombination rates to get high fitness values. How did the authors ensure that this tuning improved the predictive power instead of biasing the model even further? Under which criteria was the tuning performed?

Is there any evidence or indication that this method can be generalized to different properties?

At last, I am wondering what physical insights were gained from this work. While the method has found new materials, I am not sure how this method increased our understanding of HEAs. During the CCA section, the authors spoke of valence electron concentrations and the mixing enthalpy and entropy, but when the new materials were synthesized, these properties are never mentioned again. However, according to the CCA analysis, they are key to the hardness values. Establishing a trend could help expanding the search to different HEAs, or at least explain why some materials fall within or outside the 90% confidence interval.

Reviewer #3 (Remarks to the Author):

Materials informatics for the screening of multi-principal elements and high entropy alloys, Rickman et al.

Synopsis: The authors perform canonical correlation analysis (CCA) on published data to find factors that are significant with respect to predicting hard HEAs. Using 16 elements, they then perform a genetic algorithm (GA) on a population of 500 randomly selected specimens

(chromosomes), with an objective of achieving high hardness H while retaining some ductility. Each contains five elements, with 16 molar compositions allowed per element. The GA modifies the population towards higher H compositions. Seven of these are selected, fabricated by arc melting and H is measured.

I read the paper with interest. I have the following questions and comments.

- 1) The introduction states “there remain challenges in understanding the origin of superior properties in HEAs”. The analysis is based on statistical extractions from data, however. Can the authors present a new fundamental insight based on physical principals that derives from this work?
- 2) I did not find H values in ref. [22]. Please provide the source of these values. What is the lab-to-lab variability in H values? (What is the uncertainty in the input values to H ?)
- 3) I have a number of questions and requested clarifications about the results. I am not clear (top of p. 10) what is meant by “conservatively select a subset of alloys whose predicted values are somewhat beyond the range of the existing data set”. Does this mean that some values outside the 90% confidence limits are expected, and that the newly measured data in Fig. 2 (with apparently 5 of 7 within the bounds) confirms that? In what sense is that “conservative”? Do the predictions correctly rank the hardness of the selected alloys? What were the predicted values of the compositions versus the actual values? It seems the compositions and their predicted and measured values should be listed in a table so other workers can see if they can reproduce the results. Why do most of the selected compositions have values $V(1) \approx 1$? Is this just because that value tends to predict high H ? Were the indents carefully checked for radial cracks, thereby confirming sufficient ductility at room temperature?
- 4) Can the authors provide some useful reference numbers for H for conventional solid solutions so that a more general audience can take more meaning from the results?

Response to Referees

"Materials Informatics For the Screening of Multi-Principal Elements and High-Entropy Alloys"

We thank the reviewers for their careful reading of our manuscript and for their detailed comments/criticisms. We are pleased that they have recognized the merit and general interest in our paper. We provide a point-by-point response to these comments below and indicate the corresponding revisions made to the text to address the points raised by the reviewers. (The reviewers' comments are italicized.)

Reviewer #1

My main concern is that it is not clear why the fitness function should be determined by CCA and why CCA is helpful at all in this problem.

In an alloy design problem, one typically has some goals and constraints and those uniquely determine what the optimization problem should be. In this context, all is needed is a standard statistical regression of each target variables (say hardness) on the various "alloy metrics". No CCA is needed for this.

Thank you for this suggestion. While one might perform separate, univariate regressions, as a first approximation, to determine the dependence of hardness on the metrics, this procedure neglects correlations among both independent and dependent variables. We began by conducting a CCA that has both multiple independent variables (alloy metrics) and multiple dependent variables (hardness, phase information, etc.). The CCA incorporates, for example, correlations among dependent variables that are ignored in multiple, univariate regressions. The CCA is the most general multivariate regression and, because dependent variables correlations are embodied in the analysis, it is preferable to multiple, univariate regressions.

We then, for clarity (as our focus was on hardness), specialized to the case of one dependent variable, namely the hardness. For this special case, we are, in effect, performing a multiple regression. One can then ask, for example, whether the simple regression estimate

$$\tilde{y} = \tilde{\beta}_0 + \tilde{\beta}_1 x_1$$

is equivalent to the multiple regression estimate

$$\bar{y} = \bar{\beta}_0 + \bar{\beta}_1 x_1 + \bar{\beta}_2 x_2.$$

One finds that, in general, $\tilde{\beta}_1 \neq \bar{\beta}_1$ unless x_1 and x_2 are uncorrelated (or $\bar{\beta}_2 = 0$). Thus, the CCA and a series of univariate regression will not, in general, produce the same results, except under very special circumstances as the latter neglects correlations.

An examination of the correlation matrix, Σ , for this problem shows that the alloy metrics are, in fact, strongly correlated in some instances. For example, $\Sigma_{\text{vec}, \Delta H_{\text{mix}}} = 0.62$, $\Sigma_{\Delta H_{\text{mix}}, \text{Epar}} = -0.70$ and $\Sigma_{\delta, \text{Epar}} = 0.64$, and so the CCA approach is preferable in this context as well.

To emphasize these points, we have added the following text to our discussion of the CCA.

“CCA is preferred here to the use of multiple, univariate regressions since CCA embodies correlations among both the alloy metrics (predictor variables) and between these metrics and the output variables.”

Further down in this section after we indicate that we are focusing on the hardness, we have added the following text. In addition, we have now added a color map (new Fig. 2) of the correlation matrix that shows these correlations and motivates the use of our multivariate analysis.

“Again, multiple regression CCA is preferred here to the use of multiple, univariate regressions as the former incorporates correlations among the alloy metrics. These correlations can be seen by inspection of a color map of the correlation matrix (Fig. 2) given below.”

The CCA variate is then the natural function to use as a fitness function since one can now vary the elements in the palette and their associated compositions to systematically increase the fitness function.

Also, some of the various "alloy metrics" are fairly simple functions of composition, so it should be possible to substantially narrow down the search by performing some of the optimization analytically rather than entirely by brute force genetic algorithm (GA). It's not really a case where are a crucial part of the solution.

We agree that it may be possible to narrow the search space to some degree by analytical optimization. However, we note that some of the alloy metrics involve the logarithms of the composition variables (ΔS_{mix}), other non-linear dependence on the composition (ΔH_{mix}) and more complex ratios (Ω). For this reason, we employed a numerical solution using all of the alloy metrics rather than attempting a preliminary narrowing. For this purpose, the GA, while not absolutely critical, was convenient and produced successive generations having improved properties.

To emphasize these points, we have added the following text to the section in which we outline the GA.

“While it may be possible to narrow the search space to some degree by analytical optimization, the non-linear dependence of some of the alloy metrics on composition suggested a numerical solution using all of the metrics.”

Finally, one does not get a sense that the proposed alloys are in any sense "optimal". The statement: "Given that the GA produces an extremely fit population of high-hardness alloys that represent (in some cases) an extreme extrapolation from the $M = 82$ dataset, we conservatively select from this population a subset of alloys whose predicted hardnesses are somewhat beyond the range of the existing data set." raises some questions in that regard. (And, no, removing the

sentence is not the solution.) What would have been the best candidate materials based on the model? Why exclude it?

The reviewer raises a good point. The CCA produces a description of the data over the range of the data, and our aim is to extrapolate beyond this range. Given this inherent limitation of the CCA variates, the extrapolation should be done with some care and tested. As noted above, the GA produced successive generations having improved properties that, taken to optimality, yields candidates that are well beyond the range of the data. We have found that the optimal solution (or at least the solution found after very many GA generations) represents an extrapolation that is very far beyond the CCA range (i.e., has a fitness function that is about 300-400% greater than the maximum found with the CCA). Thus, we have used the GA to create many candidates that are somewhat beyond the CCA range (i.e., having fitness values that are about 10-15% greater than the maximum found with the CCA). Our view is that, used in this way, the GA systematically creates a large pool of potential candidates having relatively high hardnesses for a range of fitness values that represents a conservative extrapolation of the data. This extrapolation can be tested step by step and, if desired, the new alloys discovered in this way may be then added to the existing data to bootstrap the extrapolation. The goal of finding an optimal alloy in this context via an extreme extrapolation is, in our view, somewhat unrealistic. We note that a number of the candidates identified here using our approach have substantially improved hardnesses.

To clarify these points and our use of the GA in this context, the following text has been added to the manuscript in the subsection entitled “Comparison with Synthesized Multi-Principal Element Alloys”.

“Given that the GA produces an extremely fit population of high-hardness alloys that represent at optimality an extreme extrapolation from the $M=82$ dataset, we conservatively select from this population a subset of alloys whose predicted hardnesses are somewhat beyond the range of the existing data set (i.e., having fitness values that are about 10-15% greater than the maximum found with the CCA). Our view is that, used in this way, the GA systematically creates a large pool of potential candidates having relatively high hardnesses for a range of fitness values that represents a conservative extrapolation of the data. This extrapolation can be tested step by step and, if desired, the new alloys discovered in this way may be then added to the existing data to bootstrap the extrapolation.”

Overall, the authors did do a lot of work (on the experimental side) and do suggest improved alloys, so the paper does have merit. It's just a matter of motivating the methodology better or just improving it.

We thank the reviewer for this assessment and believe that the discussion above and the modifications to the manuscript will address the reviewer's concerns.

More minor points:

-Figure 1 is not helpful at all.

-Please cite where you obtained these alloy metrics from.

We believe that Fig. 1 does show important trends in the data, as noted in the manuscript. However, if the reviewer believes that it should be removed, we will certainly remove it.

In Ref. [22] there are several tables containing values for the alloy metrics. We have also now cited the paper by F. Tian *et al.* that summarizes the metrics for other alloys and have also cited papers indicated in the table in (old version) Ref. [22].

Reviewer #2

They predict seven new HEAs, two of them having hardness values above 1,000. The idea behind this method appears solid, but there are a few questions about the execution. I do not think there is sufficient data in this manuscript to justify some of their claims, and there does not appear to be sufficient additional physical insights to justify a publication in Nature Communications. I also find the necessity of tuning the GA parameters, especially without further description of how they were tuned, to be problematic. The following is a more detailed critique of their work.

We thank the reviewer for the positive assessment of our methodology. With regard to the data, we have spent considerable time searching the literature to identify HEA data with reliable hardness values. It is certainly possible, and even likely, that we missed some references in this process. Nevertheless, we believe that we have assembled a sufficiently large database for our purposes. As discussed in the paper, we can determine whether our dataset is sufficiently large to validate our conclusions by using tests of statistical significance, such as the Wilk's lambda (which is a function of data size) and Rao's F, in a hypothesis test. The small p values obtained with this analysis indicate that our assertions are justified for the dataset employed here.

With regard to the GA, we note that parameter tuning is intrinsic to this method and depends on the size and topology of the search space. There are no hard and fast rules. In practice, one observes the evolution of the fitness function over successive generations and then adjusts the parameters using heuristics so that there is substantial improvement over many generations. As discussed above in response to Reviewer #1, we use the GA to create systematically a large pool of potential candidates having relatively high, though not necessarily optimal, hardnesses for a range of fitness values. Thus, parameter adjustment is used to create this large pool, and the identification of precise values for these parameters is not necessary so long as many candidates are generated.

To clarify these points, we have added the following text to the paper below Table I in addition to the text added in response to the comments by Reviewer I (see above).

“In practice, one observes the evolution of the fitness function over successive generations and then adjusts the parameters so that there is substantial improvement over many generations. Multiple trials consisting of at least 1000 generations were used to identify superior alloys.”

Further below in the subsection entitled “Comparison with Synthesized Multi-Principal Element Alloys”, we have added the following text to the description of the GA.

“Parameter tuning in the GA is then performed to create this large pool.”

Finally, with regard to physical insights, we address this issue at the bottom of this response. In particular, we highlight the insights already given in the manuscript and a discussion of insights that have been added in response to the reviewer’s comments.

The discussion of the CCA analysis could significantly improve by being more explicit. There are a few choices that the authors make that are unclear. For example, they “conveniently” chose the reference alloy CoCrFeNiCu, but it is unclear to me what is convenient about this choice.

We agree that providing additional details here would be helpful. With regard to the reference alloy, we selected it essentially arbitrarily so that we could work with reduced units. There was no compelling reason for this choice, except perhaps that this particular alloy is relatively well studied.

To emphasize this point, we added the following text to the subsection entitled “Parallel-Coordinate Plot and Correlation Analysis” .

“This choice of reference alloy is essentially arbitrary, although we chose here a relatively well-studied alloy.”

We have also provided more details on the CCA methodology that are helpful for interpretation, along with a determination of the variability in our analysis in the Discussion section. In particular, in the Methods section we have modified and expanded the CCA description as follows.

“This is accomplished by constructing, from products of blocks of the correlation (or covariance) matrix, an operator whose eigenvalues (λ_i , where i runs from 1 to the number of variates pairs, n) capture the degree of correlation between combinations of predictor and outcome variables and whose eigenvectors determined the relative weights of the variables \cite{knapp78,rickman2017,jobson1992}. The correlation coefficient associated with a given variate i is $\sqrt{\lambda_i}$ and the associated variability is $\lambda_i / \sum_{j=1}^n \lambda_j$.”

Another such choice is the inclusion of the crystal structure properties (BCC only, FCC only, intermetallic phases) in the outcome variables. Including these variables seems especially superfluous since a) these properties are never discussed and b) in the last paragraph of the CCA analysis, they were omitted anyway. The authors should describe why these outcome variables were included in the first place and how they are relevant to the paper.

On page 7, the authors mention the correlation values of three canonical variates, but do not describe what these variates are or what is different between the variates with different

correlations.

We agree that these aspects of the analysis need clarification. We first described a CCA that had output variables (i.e., BCC only, FCC only, intermetallic phases) in addition to the hardness to demonstrate that a general analysis that links the alloy metrics with both properties and phase information is possible. We believe that this general analysis will be useful to workers who distinguish between HEAs and multi-principal element alloys based on whether the system is a solid solution. The canonical variate W is a linear combination of the hardness and the phase variables, and the three statistically relevant variates describe scenarios in which the phase variables are either positively or negatively correlated with the alloy metrics. Thus, one can identify cases in which high hardness is associated with solid solution behavior, and other cases in which high hardness is associated with the presence of intermetallic phases.

However, as stated later in the paper, our primary focus in the paper is on hardness irrespective of crystal structure and phase content. For this reason, we considered a CCA analysis with a truncated set of output variables. Nevertheless, we believe that it is valuable to outline the extended analysis presented in the paper.

To clarify these points, we have included the following text in Parallel-Coordinate Plot and Correlation Analysis subsection.

“From this investigation one can identify cases in which high hardness is associated with solid solution behavior, and other cases in which high hardness is associated with the presence of intermetallic phases. An examination of the variates indicates that the first variate (i.e., the one having the largest correlation coefficient) is most associated with the hardness and accounts for nearly 50% of the variability. The remaining variates primarily reveal associations between the alloy metrics and the phase behavior variables with only a weak dependence on hardness. Thus, this general CCA links the alloy metrics with both properties and phase information, and is especially useful to workers who distinguish between HEAs and multi-principal element alloys based on whether the system is a solid solution. We will return to this extended CCA analysis in the Discussion section below.”

First of all, the description of the currently existing literature on the prediction of HEAs needs to be improved.

We agree that our literature summary is not wholly adequate. We have therefore added many additional references, especially those describing the mechanical properties of these alloys. We have also added a paper by F. Tian *et al.* that provides a nice summary of important alloy metrics.

First of all, the method and instrument with which the hardness is determined are neither described nor cited.

We agree that more information is needed. The following text was added to the Methods section of the paper to provide this information.

“Vicker's indentation hardness testing was conducted on each as-cast alloy using a LECO LM 248AT model hardness tester. Each indent was performed on a polished surface, and a 100g load was applied for 10 seconds. At least 20 indents were conducted for each alloy using a 2x10 array, and the error was calculated to reflect 95% certainty using the Student-t distribution.”

Second, the authors present no experimental evidence that these compounds are solid solutions, which is what they wanted to find, or that they are homogeneous.

As noted above, our primary focus in the paper is on hardness irrespective of crystal structure and phase content. Our rationale for including the results of the CCA with phase information was also described above.

At last, I find their presentation of how well experiment and model agree questionable. 70% sounds very good, but it is only based on 7 data points, so the error bar is considerable. Moreover, this value can only be reached by including the data point that just barely touches the 90% confidence interval. I do not think that this is an accurate representation of the predictive power of the model without seeing more data. The authors also highlight the two compounds with a hardness value of over 1,000 in Figure 2. Interestingly, these are the values that are slightly above the 90% confidence interval. Is there a hypothesis about why that is the case? The authors should discuss this. And what is the source of the remaining outlier?

We agree that one should present the results with some care. We note that it was time-intensive to fabricate and test the samples used here, and so 7 data points represents a considerable effort. With regard to the confidence interval, a 90% interval is commonly used to express confidence, but is, of course, somewhat arbitrary. Had we used, for example, a 95% confidence interval, which is also common, data points for the fabricated samples just at the limits of the 90% band would lie within the prediction band. Thus, it is probably not useful to focus on small changes in the width of the band. For this reason, there is no hypothesis as to why the very high hardness alloys are at or above the 90% band. There is clearly some inherent scatter in the database of 82 alloys due to outliers, and it seems that this scatter leads to some outliers in the predicted data.

To emphasize these points, we have added the following text at the end of the Discussion section.

“In summary, given the number of candidate alloys considered here and the inherent scatter in the original data, this analysis provides a convenient means to predict alloys with enhanced hardness with 90% confidence.”

In short, we believe that our results are predictive and that the sample size used here is adequate as we are careful to state (now more explicitly) these predictions with caveats and with the corresponding confidence interval.

As the authors mention, the sample size is fairly small due to the limited amount of experimental data. This naturally introduces bias in the dataset. Furthermore, the authors needed to “tune” the mutation and recombination rates to get high fitness values. How did the authors ensure that

this tuning improved the predictive power instead of biasing the model even further? Under which criteria was the tuning performed?

The reviewer makes a very valid point that the dataset, owing to its size, may be biased. We certainly recognize this possibility and the fact that, given that the data comes from many different groups, the reliability of the data may vary from group to group. This is obviously a limitation of any data-intensive study. There are some safeguards, however, including checking for outliers and determining the sensitivity of the analysis to the addition of more data. Clearly it is desirable to repeat this analysis as more data become available.

With regard to tuning the GA parameters, we have addressed these concerns above as they were made earlier in the review. To avoid unwanted bias in using the GA, as mentioned in the paper we start from a randomly selected initial population chosen from our palette and repeat this for many trials. We don't believe that subsequent tuning of the recombination and mutation steps limits predictive power by somehow restricting opportunities for the GA to find fit populations.

Is there any evidence or indication that this method can be generalized to different properties?

The CCA and GA methodologies are quite general and can therefore be applied to properties other than hardness. This generalization is outlined in the Discussion section. As noted in the Introduction, a CCA and its relative, a Monte Carlo CCA, have been used in a very different context to establish correlations between ceramic powder chemistry and the resulting microstructure of a dense, sintered ceramic. A similar CCA analysis has also been used to relate the microstructure to the optoelectronic properties of thin-film solar cells (J. M. Rickman, Y. Wang, A. D. Rollett, M. P. Harmer and C. Compson, "Data Analytics using Canonical Correlation Analysis and Monte Carlo Simulation," *J. npj Computational Materials* **3**, art. no. 26 (2017)). We therefore believe that our approach is robust and applicable to a wide variety of material properties.

To emphasize these points, we have expanded the Discussion section by including the following text.

"The CCA and GA methodologies are quite general and can therefore be applied to properties other than hardness. As noted in the Introduction, a CCA and its relative, a Monte Carlo CCA, have been used in a very different context to establish correlations between ceramic powder chemistry and the resulting microstructure of a dense, sintered ceramic. A similar CCA analysis has also been used to relate the microstructure to the optoelectronic properties of thin-film solar cells."

At last, I am wondering what physical insights were gained from this work. While the method has found new materials, I am not sure how this method increased our understanding of HEAs. During the CCA section, the authors spoke of valence electron concentrations and the mixing enthalpy and entropy, but when the new materials were synthesized, these properties are never mentioned again. However, according to the CCA analysis, they are key to the hardness values. Establishing a trend could help expanding the search to different HEAs, or at least explain why some materials fall within or outside the 90% confidence interval.

We must disagree with the assessment that there is insufficient physical insight given in the paper. We note that we have discussed some aspects of the physical interpretation of our approach already in terms of the entropy of mixing, the enthalpy of mixing and the valence electron concentration in the subsection entitled “Parallel-Coordinate Plot and Correlation Analysis” with the text:

“More specifically, from Table I it can be seen that $\Delta S_{\text{mix}}^{\prime}$ are positively correlated with the H , suggesting that compositional disorder enhances the hardness. By contrast, $\Delta H_{\text{mix}}^{\prime}$ and VEC^{\prime} are negatively correlated with the hardness, suggesting in the first case that phase separation is associated with lower hardness values.”

Nevertheless, we have included additional physical interpretation. First, we note that the negative correlation of VEC with hardness found here is consistent with recent findings by Chen *et al.* (new reference) that a low VEC is associated with an improvement in strength (and therefore hardness) via the promotion of a bcc phase. To emphasize this point, we have added the following text to the “Parallel-Coordinate Plot and Correlation Analysis” subsection:

“In the latter case, the negative correlation of VEC’ with hardness is consistent with recent findings that a low VEC’ is associated with an improvement in strength via the promotion of a bcc phase.”

Next, we have now included a table (new Table I) that summarizes the standardized canonical weights found in our CCA analysis. From this table it is evident that the entropy of mixing, the enthalpy of mixing and the valence electron concentration are the most prominent metrics.

Finally, we sought to determine whether these three principal metrics embody most of the physics that determines the hardness. If so, one would expect that a CCA based only on this reduced set of variables would satisfactorily predict the hardness. To assess the adequacy of this (null) hypothesis, we conducted a CCA of this reduced model and computed the resulting F-statistic that compares the sum-of-squares errors for the reduced and full models. This statistical test permits us to assess whether including only a limited number of metrics permits an adequate description of the hardness. The resulting p-value of 0.81 suggests that the reduced model is indeed adequate. Thus, from our analysis we have identified the three most important factors dictating the hardness and provided a physical interpretation of these factors. The interplay among these factors is complex and captured by the CCA. Finally, we note that one benefit of our analysis is a dimensionality reduction that permits one to focus on a smaller set of metrics. The elimination of unimportant degrees of freedom does itself provide physical insight.

The following text was added to the manuscript to summarize these findings.

“Finally, one may ask whether the three principal metrics, $\Delta S_{\text{mix}}^{\prime}$, $\Delta H_{\text{mix}}^{\prime}$ and VEC^{\prime} , embody most of the physics that determines the hardness. If so, one would expect that a CCA based only on this reduced set of variables would satisfactorily predict the hardness. To assess the adequacy of this (null) hypothesis, we conducted a CCA of this reduced model and computed the resulting F-statistic that compares the

sum-of-squares errors for the reduced and full models \cite{mooney1999}. The resulting p-value of \$0.81\$ suggests that the reduced model is indeed adequate.”

Reviewer #3

I read the paper with interest. I have the following questions and comments.

We thank the reviewer for his/her interest in our work.

The introduction states “there remain challenges in understanding the origin of superior properties in HEAs”. The analysis is based on statistical extractions from data, however. Can the authors present a new fundamental insight based on physical principals that derives from this work?

I did not find H values in ref. [22]. Please provide the source of these values. What is the lab-to-lab variability in H values? (What is the uncertainty in the input values to H?)

We have discussed the physical insights already presented in the paper as well as those added to address the concerns of the reviewer in the response to Reviewer #2 (see above).

With regard to the H values, this was an omission on our part. Old reference 22 (Table I in this reference) listed many of the papers from which we obtained hardness values. We have now added specific references to the papers from which we obtained these values, along with another paper by F. Tian *et al.* the properties of some of these alloys. With regard to uncertainties, for the most part they were not reported in the papers that we used here. We find this to be a frustrating, but common situation. For this reason, we have conducted multiple micro-hardness tests for the alloys that we’ve synthesized and reported the associated errors.

I have a number of questions and requested clarifications about the results. I am not clear (top of p. 10) what is meant by “conservatively select a subset of alloys whose predicted values are somewhat beyond the range of the existing data set”. Does this mean that some values outside the 90% confidence limits are expected, and that the newly measured data in Fig. 2 (with apparently 5 of 7 within the bounds) confirms that? In what sense is that “conservative”? Do the predictions correctly rank the hardness of the selected alloys? What were the predicted values of the compositions versus the actual values? It seems the compositions and their predicted and measured values should be listed in a table so other workers can see if they can reproduce the results. Why do most of the selected compositions have values $V(1) \approx 1$? Is this just because that value tends to predict high H? Were the indents carefully checked for radial cracks, thereby confirming sufficient ductility at room temperature?

The reviewer raises several important questions.

With regard to the conservative selection of a subset of alloys, we have explained this in some detail in our response to Reviewer #1 above. Given the inherent scatter in the alloy data, one does expect that some hardness values will lie outside the confidence interval, as seen in the newly measured data. As can be seen from the figure (now Fig. 3), the predictions do generally

rank hardnesses correctly; however, there are deviations from predicted behavior due to the scatter in the original data.

As requested by the reviewer, we have added a table (Table II) summarizing the hardness values of the alloys synthesized here.

As stated in the manuscript, we employed 16 possible compositions per candidate element in the GA. For the purposes of fabrication, we selected compositions as close to the predicted compositions as possible for use in the arc melter.

As noted by the reviewer, most of the synthesized alloys have a variate value, $V^{(1)}$, around 1. This is because we deliberately identified alloys that have fitness values that are at, and somewhat beyond, the largest values for the alloy data set (i.e., high hardness alloys). The reasons for deliberately selecting these alloys are given in our response to the questions raised by Reviewer #1 (see above) about the use of the GA in this context.

With regard to cracking, it was found that one sample ($\text{Co}_{33}\text{W}_7\text{Al}_{33}\text{Nb}_{24}\text{Cr}_3$) exhibited cracking. However, it is difficult to know whether pre-existing cracks were present prior to indentation due to high cooling rates associated with solidification or due to a lack of ductility.

Can the authors provide some useful reference numbers for H for conventional solid solutions so that a more general audience can take more meaning from the results?

For reference, here are some hardness values for a few reference alloys.

Alloy	Hardness (HV)
Ti-5Al	532.3
Ti-5V	429.3
75Au-7Cu-13Ag-5Co (wt %)	128

We have incorporated this information in the References section of the manuscript with the following text.

“For comparison, the Vickers hardness values (HV) for alloys such as Ti-5Al, Ti-5V and 75Au-7Cu-13Ag-5Co (wt. %) are 532.3, 429.3 and 128, respectively.

Lim, H-S. *et al.*. Evaluation of Surface Mechanical Properties and Grindability of Binary Ti Alloys Containing 5 wt % Al, Cr, Sn, and V. *Metals* **7**, 487 (2017).

Suss, R. *et al.*. 18 Carat Yellow Gold Alloys with Increased Hardness. *Gold Bulletin* **37**, 3-4 (2004).”

Reviewers' comments:

Reviewer #1 (Remarks to the Author):

My comment was NOT about contrasting CCA and univariate regressions (the latter would have been obviously completely efficient and was clearly not what I suggested - "regression" was singular in my report).

My point was to contrast CCA and a single multivariate regression that relates hardness to all the "alloy metrics".

The authors' answer now clearly indicates that they are, in fact, simply using a multivariate regression and not CCA. The whole discussion about CCA is completely superfluous and actually will confuse readers.

The authors should just re-write the paper to indicate that they are using a regression. Period. Apologies if this makes the paper sound less avant-garde...

The explanation for not exploring the best predicted alloy predicted by their statistical analysis is still neither very clear nor satisfying. It should be possible to give rather formal answer to this. The authors have all the standard errors on the regression parameters and the variance of the residuals, so they could calculate a very reasonable estimate of the prediction error of the model outside the range where it was fitted. One can then limit the search to regions where that predicted error is not too large.

On a similar note, the confidence bands in Figure 3 seem incorrect: They should widen towards the left and towards the right if they properly account for the variance in the estimated coefficients.

This effect is what would allow the authors to restrict their extrapolation to where the model has a good predictive power.

The color map (Fig 2) of the correlations is not helpful if one does not know which color '0' corresponds to.

Figure 1 is as unhelpful as before.

Reviewer #2 (Remarks to the Author):

I would like to thank the authors for their thorough response to my review. They sufficiently addressed my questions, and I think the paper in its current form can be published. I only have a few comments that the authors may want to consider.

The authors wrote in their response:

"With regard to the reference alloy, we selected it essentially arbitrarily so that we could work with reduced units. There was no compelling reason for this choice, except perhaps that this particular alloy is relatively well studied."

I think using a well-studied alloy as a reference is a very compelling reason and could be emphasized over the arbitrariness of the choice.

I would also like to clarify that my comment with regards to the presentation of how well experiment and model agree were by no means intended to criticize the size of the data set. I have no doubt that there was considerable effort behind generating it, especially given the limited experimental data the authors had access to. My problem was the presentation of the results ("70%" vs. "five out of seven"), not the handling of the data.

At last, I would like to address the experimental work. It may not be strictly necessary for the

broader point of the paper to determine the crystal structure/phase behavior of the synthesized alloys. However, I think taking x-ray diffraction patterns and presenting the results would round out the paper nicely, especially in light of their comment on page 14: "The methodology summarized here can, of course, be employed to examine the influence of the predictor set on hardness and phase behavior, as summarized in Eq. (1). This analysis is useful if one wishes to obtain hard alloys that are also likely to be a single solid solution."

Reviewer #3 (Remarks to the Author):

The authors assert that a p-value of 0.81 among a reduced set of the the mixing entropy, the mixing enthalpy and the VEC provides physical insight through the elimination of unimportant degrees of freedom. This may provide a moderate statistical insight ($p=0.81$ is below 90% confidence, which is generally the minimum for a truly significant effect in experiments). The authors associate positive (ΔS , with disorder) and negative (ΔH , with phase separation) (VEC, view promotion of the BCC phase) correlations with high H. While these are plausible, apparently none are demonstrated via microstructural analysis. As they have not investigated whether the resulting materials are hardened by a solutionizing effect or by a precipitate mechanism (see response to Reviewer #2), this further indicates that there is statistical, but not verified physical insight. If the authors make a statement regarding further substantiation is necessary to confirm the guidance they provide (in terms of microstructural and perhaps computational analysis), I feel the statements on physical insight would provide better perspective to the reader.

Table III now provides measured H values. What are the predicted values? Please provide both.

It would seem straightforward to determine if the observed cracking in (Co 33 W 7 Al 33 Nb 24 Cr 3) occurred after processing or after indentation. If it occurred after processing alone, how does this square with the authors' assertion of high ductility for all the alloys? If it occurred after indentation, again how can this be reconciled in view of the authors' assertion of high ductility for all the alloys?

Response to Referees

"Materials Informatics For the Screening of Multi-Principal Elements and High-Entropy Alloys"

We thank the reviewers for their careful reading of our revised manuscript and for their detailed comments/criticisms. We believe that our paper has been substantially improved in this process. With regard to the remaining comments, we provide a point-by-point response below and indicate the corresponding revisions made to the text to address the points raised by the reviewers. (The reviewers' comments are italicized.) Changes in the manuscript are highlighted in yellow.

Reviewer #1

My comment was NOT about contrasting CCA and univariate regressions (the latter would have been obviously completely efficient and was clearly not what I suggested - "regression" was singular in my report). My point was to contrast CCA and a single multivariate regression that relates hardness to all the "alloy metrics". The authors' answer now clearly indicates that they are, in fact, simply using a multivariate regression and not CCA. The whole discussion about CCA is completely superfluous and actually will confuse readers. The authors should just re-write the paper to indicate that they are using a regression. Period. Apologies if this makes the paper sound less avant-garde...

We take the reviewer's point. For the main part of the analysis we did use a multiple regression, the limiting case of a CCA with one output (the hardness). We thought that we had emphasized this point with the sentence "Given that our primary focus here is on hardness irrespective of crystal structure and phase content, we consider next a CCA analysis with a truncated set \vec{y} consisting of only H^{\prime} " and our use of the term "multiple regression CCA". However, to avoid confusion, we have now rewritten the beginning of the subsection Parallel-Coordinate Plot and Correlation Analysis to emphasize that the analysis is based on a multiple regression. The revised text is given below.

"To be more quantitative, we employ a multiple regression analysis to assess the relative importance of various materials characteristics whose complex interplay dictates properties. Our analysis begins with a vector of predictor variables, \vec{x} , and outcome variables, \vec{y} , taken as
$$\begin{eqnarray} \label{eq:inputoutput1} \vec{x} = \{\Delta^{\prime}, \Delta H_{\text{mix}}^{\prime}, \Delta S_{\text{mix}}^{\prime}, T_m^{\prime}, \Omega^{\prime}, \mathcal{E}^{\prime}, \text{VEC}^{\prime}\}, \nonumber \\ \vec{y} = \{H^{\prime}\}, \end{eqnarray}$$

where \vec{y} consists of only H^{\prime} for a multiple regression. (A more general CCA analysis that is useful when there are additional outcome variables in \vec{y} is described below in the Discussion section.) It should be noted that correlations, some of which are significant, exist among the predictor variables as can be seen by inspection of a color map of the correlation matrix (Fig. \ref{fig:corrmatrixplot}) given below."

As the full CCA analysis is useful if one wishes to also perform, for example, a phase analysis (the subject of ongoing work) we would prefer to leave some discussion of this option in the paper. To avoid confusion, we are placing this text, along with the results of this analysis, later in the Discussion section of the paper where we outline future work. Thus, the discussion of the central analysis is now based on multiple regression and the GA.

The explanation for not exploring the best predicted alloy predicted by their statistical analysis is still neither very clear nor satisfying. It should be possible to give rather formal answer to this. The authors have all the standard errors on the regression parameters and the variance of the residuals, so they could calculate a very reasonable estimate of the prediction error of the model outside the range where it was fitted.

This is a good point. Using the hardness values obtained for the synthesized samples, we computed the standard error of the estimate and obtained a value of 259.2 HV. We modified the paper by including the following text.

“The standard error of the estimate, as obtained using the data for the synthesized alloys, is a measure of the prediction error and is approximately 259.2 HV.”

On a similar note, the confidence bands in Figure 3 seem incorrect: They should widen towards the left and towards the right if they properly account for the variance in the estimated coefficients. This effect is what would allow the authors to restrict their extrapolation to where the model has a good predictive power.

With regard to the confidence bands, we used a prediction band corresponding to a single observation, rather than one associated with the mean. The former incorporates information on variations in the parameters and the responses while the latter only incorporates information on parameter variations. This latter type of band (as noted by the reviewer) generally shows greater variability in width. The single-observation prediction band seemed more appropriate here as we wished to convey something about the confidence in the prediction of single observations. Moreover, the prediction band based on the mean would underestimate this error. To emphasize this point, we have indicated that the prediction band that we are using is a single-observation band in the caption for Fig. 3.

The color map (Fig 2) of the correlations is not helpful if one does not know which color '0' corresponds to. Figure 1 is as unhelpful as before.

We would prefer to keep Fig. 1 as we feel it does exhibit important trends. For the color map, the color zero corresponds to white. We have modified the figure caption by adding the following sentence:

“No correlation (0) corresponds to white.”

Reviewer #2

I would like to thank the authors for their thorough response to my review. They sufficiently addressed my questions, and I think the paper in its current form can be published. I only have a few comments that the authors may want to consider.

We thank the reviewer for the very positive assessment of our work.

“With regard to the reference alloy, we selected it essentially arbitrarily so that we could work with reduced units. There was no compelling reason for this choice, except perhaps that this particular alloy is relatively well studied.”

I think using a well-studied alloy as a reference is a very compelling reason and could be emphasized over the arbitrariness of the choice.

We agree with the reviewer and have modified the text as follows to stress that the reason for the choice of the reference alloy is that it is well-studied.

“This choice was made given that the reference alloy is relatively well-studied.”

I would also like to clarify that my comment with regards to the presentation of how well experiment and model agree were by no means intended to criticize the size of the data set. I have no doubt that there was considerable effort behind generating it, especially given the limited experimental data the authors had access to. My problem was the presentation of the results (“70%” vs. “five out of seven”), not the handling of the data.

Thank you for clarifying this point. We have modified the text to indicate that about five out of the seven candidates are within the confidence interval, as given below.

“It can be seen from the figure that about five out of seven of the hardnesses are within”

At last, I would like to address the experimental work. It may not be strictly necessary for the broader point of the paper to determine the crystal structure/phase behavior of the synthesized alloys. However, I think taking x-ray diffraction patterns and presenting the results would round out the paper nicely, especially in light of their comment on page 14: “The methodology summarized here can, of course, be employed to examine the influence of the predictor set on hardness and phase behavior, as summarized in Eq. (1). This analysis is useful if one wishes to obtain hard alloys that are also likely to be a single solid solution.”

We appreciate the reviewer’s suggestion here. Given our focus in this paper, we believe that the x-ray work is somewhat beyond the scope of this work. However, we are continuing to work on this problem and believe that an x-ray study would fit nicely within the scope of an upcoming publication.

Reviewer #3

The authors assert that a p-value of 0.81 among a reduced set of the the mixing entropy, the mixing enthalpy and the VEC provides physical insight through the elimination of unimportant degrees of freedom. This may provide a moderate statistical insight (p=0.81 is below 90% confidence, which is generally the minimum for a truly significant effect in experiments). The authors associate positive (ΔS , with disorder) and negative (ΔH , with phase separation) (VEC, view promotion of the BCC phase) correlations with high H. While these are plausible, apparently none are demonstrated via microstructural analysis. As they have not investigated whether the resulting materials are hardened by a solutionizing effect or by a precipitate mechanism (see response to Reviewer #2), this further indicates that there is statistical, but not verified physical insight. If the authors make a statement regarding further substantiation is necessary to confirm the guidance they provide (in terms of microstructural and perhaps computational analysis), I feel the statements on physical insight would provide better perspective to the reader.

We agree that we should state that further microstructural (and possibly computational) analysis is needed to confirm the statistical insights provided in the manuscript. We have therefore added the following text to the manuscript to make this point.

“It should be emphasized though that further microstructural (and possibly computational) analysis is needed to confirm these statistical insights.”

It would seem straightforward to determine if the observed cracking in (Co 33 W 7 Al 33 Nb 24 Cr 3) occurred after processing or after indentation. If it occurred after processing alone, how does this square with the authors' assertion of high ductility for all the alloys? If it occurred after indentation, again how can this be reconciled in view of the authors' assertion of high ductility for all the alloys?

We have carefully examined the sample in question and, unfortunately, the determination as to whether cracking occurred after processing or indentation is not straightforward. Prior to the indentation tests, some cracking was observed in the as-cast alloy. Moreover, upon inspection of the test samples, it was also the case that microhardness indentations showed signs of cracks at the corners of the indentations. We include here an image illustrating this latter type of cracking.

Given this somewhat ambiguous situation, it is probably fair to say that, in addition to cracking caused by rapid cooling, there was also some sample brittleness. The Pugh criterion is a widely-used empirical criterion that correlates ductility with the ratio μ/B , as noted in the paper, and is useful in the absence of tensile-test data. Given its empirical nature, it is perhaps not surprising that it does not work in every case, such as this one. To emphasize this point, we have added the following sentence to the paper.

“However, for the hardest alloy a close inspection of the hardness indentations did reveal some degree of brittleness as cracking occurred near indentation corners in addition to observable cooling cracks.”

In ongoing work, we are investigating the strengthening mechanisms for these and other alloys. Preliminary results suggest that some strengthening may be due to the presence of dendritic and/or cellular structures, as has been observed using microscopy on a few samples, in addition to solid-solution strengthening. An image showing such structures is given below.

Response to Referees

"Materials Informatics For the Screening of Multi-Principal Elements and High-Entropy Alloys"

We thank the reviewers for their careful reading of our revised manuscript and for their detailed comments/criticisms. We believe that our paper has been substantially improved in this process. With regard to the remaining comments, we provide a point-by-point response below and indicate the corresponding revisions made to the text to address the points raised by the reviewers. (The reviewers' comments are italicized.) Changes in the manuscript are highlighted in yellow.

Reviewer #1

My comment was NOT about contrasting CCA and univariate regressions (the latter would have been obviously completely efficient and was clearly not what I suggested - "regression" was singular in my report). My point was to contrast CCA and a single multivariate regression that relates hardness to all the "alloy metrics". The authors' answer now clearly indicates that they are, in fact, simply using a multivariate regression and not CCA. The whole discussion about CCA is completely superfluous and actually will confuse readers. The authors should just re-write the paper to indicate that they are using a regression. Period. Apologies if this makes the paper sound less avant-garde...

We take the reviewer's point. For the main part of the analysis we did use a multiple regression, the limiting case of a CCA with one output (the hardness). We thought that we had emphasized this point with the sentence "Given that our primary focus here is on hardness irrespective of crystal structure and phase content, we consider next a CCA analysis with a truncated set \vec{y} consisting of only H^{\prime} " and our use of the term "multiple regression CCA". However, to avoid confusion, we have now rewritten the beginning of the subsection Parallel-Coordinate Plot and Correlation Analysis to emphasize that the analysis is based on a multiple regression. The revised text is given below.

"To be more quantitative, we employ a multiple regression analysis to assess the relative importance of various materials characteristics whose complex interplay dictates properties. Our analysis begins with a vector of predictor variables, \vec{x} , and outcome variables, \vec{y} , taken as
$$\begin{eqnarray} \label{eq:inputoutput1} \vec{x} = \{ \Delta^{\prime}, \Delta H_{\text{mix}}^{\prime}, \Delta S_{\text{mix}}^{\prime}, T_m^{\prime}, \Omega^{\prime}, \mathcal{E}^{\prime}, \text{VEC}^{\prime} \}, \nonumber \\ \vec{y} = \{ H^{\prime} \}, \end{eqnarray}$$

where \vec{y} consists of only H^{\prime} for a multiple regression. (A more general CCA analysis that is useful when there are additional outcome variables in \vec{y} is described below in the Discussion section.) It should be noted that correlations, some of which are significant, exist among the predictor variables as can be seen by inspection of a color map of the correlation matrix (Fig. \ref{fig:corrmatrixplot}) given below."

As the full CCA analysis is useful if one wishes to also perform, for example, a phase analysis (the subject of ongoing work) we would prefer to leave some discussion of this option in the paper. To avoid confusion, we are placing this text, along with the results of this analysis, later in the Discussion section of the paper where we outline future work. Thus, the discussion of the central analysis is now based on multiple regression and the GA.

The explanation for not exploring the best predicted alloy predicted by their statistical analysis is still neither very clear nor satisfying. It should be possible to give rather formal answer to this. The authors have all the standard errors on the regression parameters and the variance of the residuals, so they could calculate a very reasonable estimate of the prediction error of the model outside the range where it was fitted.

This is a good point. Using the hardness values obtained for the synthesized samples, we computed the standard error of the estimate and obtained a value of 259.2 HV. We modified the paper by including the following text.

“The standard error of the estimate, as obtained using the data for the synthesized alloys, is a measure of the prediction error and is approximately 259.2 HV.”

On a similar note, the confidence bands in Figure 3 seem incorrect: They should widen towards the left and towards the right if they properly account for the variance in the estimated coefficients. This effect is what would allow the authors to restrict their extrapolation to where the model has a good predictive power.

With regard to the confidence bands, we used a prediction band corresponding to a single observation, rather than one associated with the mean. The former incorporates information on variations in the parameters and the responses while the latter only incorporates information on parameter variations. This latter type of band (as noted by the reviewer) generally shows greater variability in width. The single-observation prediction band seemed more appropriate here as we wished to convey something about the confidence in the prediction of single observations. Moreover, the prediction band based on the mean would underestimate this error. To emphasize this point, we have indicated that the prediction band that we are using is a single-observation band in the caption for Fig. 3.

The color map (Fig 2) of the correlations is not helpful if one does not know which color '0' corresponds to. Figure 1 is as unhelpful as before.

We would prefer to keep Fig. 1 as we feel it does exhibit important trends. For the color map, the color zero corresponds to white. We have modified the figure caption by adding the following sentence:

“No correlation (0) corresponds to white.”

Reviewer #2

I would like to thank the authors for their thorough response to my review. They sufficiently addressed my questions, and I think the paper in its current form can be published. I only have a few comments that the authors may want to consider.

We thank the reviewer for the very positive assessment of our work.

“With regard to the reference alloy, we selected it essentially arbitrarily so that we could work with reduced units. There was no compelling reason for this choice, except perhaps that this particular alloy is relatively well studied.”

I think using a well-studied alloy as a reference is a very compelling reason and could be emphasized over the arbitrariness of the choice.

We agree with the reviewer and have modified the text as follows to stress that the reason for the choice of the reference alloy is that it is well-studied.

“This choice was made given that the reference alloy is relatively well-studied.”

I would also like to clarify that my comment with regards to the presentation of how well experiment and model agree were by no means intended to criticize the size of the data set. I have no doubt that there was considerable effort behind generating it, especially given the limited experimental data the authors had access to. My problem was the presentation of the results (“70%” vs. “five out of seven”), not the handling of the data.

Thank you for clarifying this point. We have modified the text to indicate that about five out of the seven candidates are within the confidence interval, as given below.

“It can be seen from the figure that about five out of seven of the hardnesses are within”

At last, I would like to address the experimental work. It may not be strictly necessary for the broader point of the paper to determine the crystal structure/phase behavior of the synthesized alloys. However, I think taking x-ray diffraction patterns and presenting the results would round out the paper nicely, especially in light of their comment on page 14: “The methodology summarized here can, of course, be employed to examine the influence of the predictor set on hardness and phase behavior, as summarized in Eq. (1). This analysis is useful if one wishes to obtain hard alloys that are also likely to be a single solid solution.”

We appreciate the reviewer’s suggestion here. Given our focus in this paper, we believe that the x-ray work is somewhat beyond the scope of this work. However, we have performed additional microstructural and microchemical analyses (see response to Reviewer #3 below) to gain further insight into the hardening mechanisms. In particular, we have used X-ray energy-dispersive spectroscopy (EDS) to construct compositional maps of the constituent elements for the synthesized alloys having the highest and lowest hardness values, and have used this information

to infer a solid solution strengthening mechanism. The manuscript has been modified to discuss these points, as described below.

Reviewer #3

The authors assert that a p-value of 0.81 among a reduced set of the the mixing entropy, the mixing enthalpy and the VEC provides physical insight through the elimination of unimportant degrees of freedom. This may provide a moderate statistical insight (p=0.81 is below 90% confidence, which is generally the minimum for a truly significant effect in experiments). The authors associate positive (ΔS , with disorder) and negative (ΔH , with phase separation) (VEC, view promotion of the BCC phase) correlations with high H. While these are plausible, apparently none are demonstrated via microstructural analysis. As they have not investigated whether the resulting materials are hardened by a solutionizing effect or by a precipitate mechanism (see response to Reviewer #2), this further indicates that there is statistical, but not verified physical insight. If the authors make a statement regarding further substantiation is necessary to confirm the guidance they provide (in terms of microstructural and perhaps computational analysis), I feel the statements on physical insight would provide better perspective to the reader.

We agree that we should state that further microstructural (and possibly computational) analysis is needed to confirm the statistical insights provided in the manuscript. We have therefore added the following text to the manuscript to make this point.

“It should be emphasized though that further microstructural (and possibly computational) analysis is needed to confirm these statistical insights.”

It would seem straightforward to determine if the observed cracking in (Co₃₃W₇Al₃₃Nb₂₄Cr₃) occurred after processing or after indentation. If it occurred after processing alone, how does this square with the authors' assertion of high ductility for all the alloys? If it occurred after indentation, again how can this be reconciled in view of the authors' assertion of high ductility for all the alloys?

This is a valid point. To understand better the observed plastic response (i.e., hardness and ductility) of the synthesized alloys, we have performed an extensive microstructural and microchemical analysis of two of the alloys, namely the ones having the highest and lowest hardness values. In particular, the microstructures of the samples were examined using scanning-electron microscopy (SEM) and the associated compositional maps of the constituent elements was obtained by X-ray energy-dispersive spectroscopy (EDS). From a comparison of these microstructures and the spatial distribution of elements in the maps we were able to infer a plausible hardening mechanism. The manuscript was modified by the addition of a section entitled “Microstructural and Microchemical Characterization” to discuss these analyses as follows:

“To understand better the observed plastic response of the synthesized alloys, the microstructure of the samples with compositions Co₃₃W₀₇Al₃₃Nb₂₄Cr₀₃ and Ti₃₉W₀₄Nb₃₁Ta₀₄Co₂₂, the alloys having the highest and lowest hardness values (see Table III), respectively, were examined using scanning-electron

microscopy (SEM). Figure 5 depicts the microstructure of $\text{Co}_{33}\text{W}_7\text{Al}_{33}\text{Nb}_{24}\text{Cr}_3$ using secondary electron contrast, as well as the associated compositional maps of the constituent elements obtained by X-ray energy-dispersive spectroscopy (EDS). It can be seen that the microstructure is dendritic (which is consistent with a cast alloy), with a relatively small volume fraction (~ 13 vol. %) of inter-dendritic phase. The EDS data shows that the dendritic phase is enriched in Nb and W. Given that there is no obvious partitioning of the Nb and W, it is assumed that they form a solid solution. This behavior would not be unexpected given that both Nb and W are BCC and exhibit complete solid solubility [Yli2013]. The SEM also reveals that the inter-dendritic phase exhibits a two-phase, eutectic-like morphology. Of the two phases, one is Nb rich; whereas the other contains higher proportions of Al, Co and Cr, with little Nb. Unlike for the case of the dendrite body, it does not appear that there is preferential association of the W with the Nb-rich component.

The corresponding images for the softest alloy are shown in Figure 6. This microstructure of $\text{Ti}_{39}\text{W}_4\text{Nb}_{31}\text{Ta}_4\text{Co}_{22}$ is also dendritic, but the volume fraction of inter-dendritic material is much higher here (~ 53 vol. %). In this alloy, the dendrites are enriched in Nb, Ta and W, whereas the inter-dendritic regions contain a greater proportion of Co and Ti. In the SEM image (Fig. 6a), the slight differences in contrast at the edges of the dendrites suggest the possibility of coring. This observation is confirmed by the compositional maps where it can be seen that for a given dendrite structure, the spatial extent of Nb enrichment is the greatest, W is confined to the dendrite inner core and the spatial extent of Ta is intermediate between the two. Thus, one infers that, although all three elements are present in the dendritic regions, the distribution is inhomogeneous.

The marked difference in hardness between the two alloys is very interesting, particularly given that the compositional make-ups of the dendritic phases are not that dissimilar. In both cases there is a significant component of Nb and W. In the case of the softest alloy, Ta is also present. One possible explanation is that the high hardness of $\text{Co}_{33}\text{W}_7\text{Al}_{33}\text{Nb}_{24}\text{Cr}_3$ is attributable to the significant volume fraction of the Nb-W phase which is solid solution strengthened. One could suggest that the same degree of strengthening is not present in $\text{Ti}_{39}\text{W}_4\text{Nb}_{31}\text{Ta}_4\text{Co}_{22}$ either due to the non-uniform distribution of the elements, or the additional presence of Ta. Of course the potential contribution of alloying elements present at relatively low concentrations cannot be excluded.”

Additional figures and references were also included in this section to show the relevant micrographs and compositional maps.

To understand better the ductility of the alloys, we performed optical microscopy to investigate the cracking behavior for the 7 synthesized alloys. This involved an examination of the 20 microhardness indentations per sample to determine what percentage of the indentations showed signs of cracks at the corners. (An example of an image showing cracking at an indentation is shown below.) We then characterized the ductility of a given alloy as follows:

Very ductile ($\geq 95\%$ of the indentations without cracks)

Moderate to considerable ductility ($\geq 50\%$ of the indentations without cracks)

Limited ductility ($\approx 15\text{-}20\%$ of the indentations without cracks)

Brittle ($\leq 10\%$ of the indentations without cracks)

As described now in the text, 2 of the 7 systems showed moderate to considerable ductility, including the relatively hard alloys $\text{Ti}_{32}\text{Nb}_9\text{Ta}_1\text{Cr}_{19}\text{Co}_{39}$ (HV 856) and $\text{Co}_6\text{W}_9\text{Al}_{36}\text{Mo}_{38}\text{Ni}_{11}$ (HV 725), while the hard alloy $\text{Ti}_{18}\text{Ni}_{24}\text{Ta}_{12}\text{Cr}_{22}\text{Co}_{24}$ (HV 1011) showed limited ductility. Moreover, the relatively low hardness alloys, $\text{Ti}_{39}\text{W}_4\text{Nb}_{31}\text{Ta}_4\text{Co}_{22}$ and $\text{Ti}_{44}\text{Ni}_2\text{Nb}_{21}\text{Cr}_{21}\text{Co}_{12}$, were found to be quite ductile, which is generally consistent with observations that Group IV element additions, such as Ti, are often associated with increased ductility. The two remaining alloys were found to be brittle. Given the empirical nature of the Pugh criterion, it is perhaps unsurprising that it is a rough predictor of ductility in this context. Nevertheless, with this methodology we have identified several alloys that are both hard and relatively ductile. To describe these results, the following text was added in the “Microstructural and Microchemical Characterization”:

“Finally, we have also assessed the ductility of each synthesized alloys by optical microscopic inspection of the (20) microhardness indentations per alloy for signs of cracks at the corners of the indentations. It was found that 2 of the 7 systems showed moderate to considerable ductility (i.e., $\approx 50\%$ of the indentations without cracks), including the relatively hard alloys $\text{Ti}_{32}\text{Nb}_9\text{Ta}_1\text{Cr}_{19}\text{Co}_{39}$ and $\text{Co}_6\text{W}_9\text{Al}_{36}\text{Mo}_{38}\text{Ni}_{11}$, while the hard alloy $\text{Ti}_{18}\text{Ni}_{24}\text{Ta}_{12}\text{Cr}_{22}\text{Co}_{24}$ showed limited ductility (i.e., $\approx 15 - 20\%$ of the indentations without cracks). In addition, the relatively low hardness alloys, $\text{Ti}_{39}\text{W}_4\text{Nb}_{31}\text{Ta}_4\text{Co}_{22}$ and $\text{Ti}_{44}\text{Ni}_2\text{Nb}_{21}\text{Cr}_{21}\text{Co}_{12}$, were found to be quite ductile (i.e., $\approx 95\%$ of the indentations without cracks), which is generally consistent with observations that Group IV element additions, such as Ti, are often associated with increased ductility *\cite{senkovmiracle}*. The remaining alloys were found to be brittle. Given the empirical nature of the Pugh criterion, it is perhaps unsurprising that it is a rough predictor of ductility in this context. Nevertheless, with this methodology we have identified several alloys that are both hard and relatively ductile.”

Reviewers' comments:

Reviewer #1 (Remarks to the Author):

The discussion starting with "Finally, to illustrate the use of the CCA..." is almost completely uninformative. It looks like just an excuse to mention CCA somewhere...

Once again, the confidence bands in Figure 3 seem incorrect. The "single observation confidence bands" must account for two effects: the uncertainty in the estimated regression coefficients AND the residual error (i.e. components not predicted by the regression). The second term can be assumed constant but the first one is not constant. So the upper and lower limits of the band in fig 3 should not be parallel, unless the uncertainty in the regression coefficients happens to be negligible.

Finally, the authors report various correlations throughout the text but no standard errors is ever given. It is thus difficult to judge the significance of such correlations.

Reviewer #2 (Remarks to the Author):

the authors have satisfied my comments and the article is good.

I recommend acceptance.

Reviewer #3 (Remarks to the Author):

It looks like the authors addressed my concerns substantially, but for the life of me I cannot read the yellow on white background. Please resubmit, with updates perhaps in red, green or blue.

We again thank the reviewers for their critical reading of our manuscript, their positive assessment and their helpful comments. As Reviewers 2 and 3 are fully satisfied with our revisions, we focus here on the remaining comments from Reviewer 1. A point-by-point response to these comments is given below, with the reviewer's comments in italics.

Reviewer #1

Once again, the confidence bands in Figure 3 seem incorrect. The "single observation confidence bands" must account for two effects: the uncertainty in the estimated regression coefficients AND the residual error (i.e. components not predicted by the regression). The second term can be assumed constant but the first one is not constant. So the upper and lower limits of the band in fig 3 should not be parallel, unless the uncertainty in the regression coefficients happens to be negligible.

We thank the reviewer for commenting on the analysis. The issue here seems to be one of terminology, and we apologize for any confusion. As noted in the caption to Fig. 3, the shaded region represents a single-observation *prediction* band. A prediction band gives the error for an output "y" value given a predictor "x" value. By contrast, a confidence band reflects the error associated with the mean of "y". The expressions for these intervals (given below) are similar, but with an important difference.

Prediction Band

$$\hat{y} \pm t_{\left(\frac{\alpha}{2}, n-2\right)} \sqrt{MSE \cdot \left(1 + \frac{1}{n} + \frac{(x-\bar{x})}{\sum_i (x_i - \bar{x})^2}\right)}$$

Confidence Band

$$\hat{y} \pm t_{\left(\frac{\alpha}{2}, n-2\right)} \sqrt{MSE \cdot \left(\frac{1}{n} + \frac{(x - \bar{x})}{\sum_i (x_i - \bar{x})^2}\right)}$$

In these expressions, \hat{y} is the predicted value associated with the predictor, x , MSE is the mean squared error, $t_{\left(\frac{\alpha}{2}, n-2\right)}$ is the value of the relevant t-statistic and n is the number of data points. It should be noted that the prediction band has an extra term, namely MSE , under the radical. In our analysis, this term dominates the others.

To illustrate the difference in these quantities, we have plotted below the prediction band and the confidence band for our data. For clarity, we have omitted the data points and just reproduced the regression line.

As can be seen from the figure, the dominance of the extra MSE term in the expression for the prediction band results in nearly linear behavior for the band limits, which are necessarily wider than the confidence bands. By contrast, the confidence band limits vary to some degree over the length of the curve, as noted by the reviewer. Since we wish to convey the uncertainty in a new value of the output variate (i.e., the standardized hardness), the prediction band was used.

Unfortunately, we referred (sloppily) once elsewhere in the text to “the pictured 90% confidence interval”. We have now replaced “confidence” by “prediction” to clarify what we have done.

Finally, the authors report various correlations throughout the text but no standard errors is ever given. It is thus difficult to judge the significance of such correlations.

We thank the reviewer for pointing this out. We have now included these standard errors in the appropriate locations in the text.

Finally, we have indicated modifications to the manuscript in red. We removed yellow highlighting and apologize for using this color in our earlier submission as it may have caused some difficulties in reading the paper.

REVIEWERS' COMMENTS:

Reviewer #1 (Remarks to the Author):

The responses to my comments are satisfactory.

(There is a typo in the response in the expression for the confidence and prediction bands: $(x - \bar{x})$ should be squared. I assume the actual calculations were done correctly, since the bands widen a bit on both ends.)

I do recognize that a lot of work went into collecting the data and the study did deliver new promising compositions.

I just wanted to make sure the statistical analysis is sound and uses the appropriate methods. There is so much misuse and misunderstanding about statistics in our field...

Reviewer #3 (Remarks to the Author):

My main comments from previous review were:

1) It would seem straightforward to determine if the observed cracking in (Co 33 W 7 Al 33 Nb 24 Cr 3) occurred after processing or after indentation. If it occurred after processing alone, how does this square with the authors' assertion of high ductility for all the alloys? If it occurred after indentation, again how can this be reconciled in view of the authors' assertion of high ductility for all the alloys?

The response regarding the hardness and microstructure is now very satisfactory.

2) Table III now provides measured H values. What are the predicted values? Please provide both.

Please add a column to Table 3 indicating the predicted hardness values in addition to the measured ones.

We again thank the reviewers for their critical reading of our manuscript, their positive assessment and their helpful comments. We address here the remaining issue raised by Reviewer #3. A response to this issue is given below, with the reviewer's comments in italics.

Reviewer #3

Table III now provides measured H values. What are the predicted values? Please provide both. Please add a column to Table 3 indicating the predicted hardness values in addition to the measured ones.

We have now computed the predicted hardness values and added these values to the table (which is now Table 2 as we were instructed to remove an earlier table from the paper.)